# A Review on Environmental Efficiency Evaluation of New Energy Vehicles Using Life Cycle Analysis

**Nenming Wang** [1,2,3,*] **and Guwen Tang** [1,2,3]

1    School of Management, Xi'an Jiaotong University, No. 28 Xianning Road, Xi'an 710049, China; tgwgb1123@stu.xjtu.edu.cn
2    The Key Lab of the Ministry of Education for Process Control & Efficiency Engineering, No. 28 Xianning Road, Xi'an 710049, China
3    ERC for Process Mining of Manufacturing Services in Shaanxi Province, No. 28 Xianning Road, Xi'an 710049, China
*    Correspondence: wangnm@mail.xjtu.edu.cn

**Abstract:** New energy vehicles (NEVs), especially electric vehicles (EVs), address the important task of reducing the greenhouse effect. It is particularly important to measure the environmental efficiency of new energy vehicles, and the life cycle analysis (LCA) model provides a comprehensive evaluation method of environmental efficiency. To provide researchers with knowledge regarding the research trends of LCA in NEVs, a total of 282 related studies were counted from the Web of Science database and analyzed regarding their research contents, research preferences, and research trends. The conclusion drawn from this research is that the stages of energy resource extraction and collection, carrier production and energy transportation, maintenance, and replacement are not considered to be research links. The stages of material, equipment, and car transportation and operation equipment settling, and forms of use need to be considered in future research. Hydrogen fuel cell electric vehicles (HFCEVs), vehicle type classification, the water footprint, battery recovery and reuse, and battery aging are the focus of further research, and comprehensive evaluation combined with more evaluation methods is the direction needed for the optimization of LCA. According to the results of this study regarding EV and hybrid power vehicles (including plug-in hybrid electric vehicles (PHEV), fuel-cell electric vehicles (FCEV), hybrid electric vehicles (HEV), and extended range electric vehicles (EREV)), well-to-wheel (WTW) average carbon dioxide ($CO_2$) emissions have been less than those in the same period of gasoline internal combustion engine vehicles (GICEV). However, EV and hybrid electric vehicle production $CO_2$ emissions have been greater than those during the same period of GICEV and the total $CO_2$ emissions of EV have been less than during the same period of GICEV.

**Keywords:** life cycle analysis; new energy vehicle; environmental efficiency

## 1. Introduction

Since 2000, $CO_2$ emissions from the global transport sector have been on the rise, from 5.8 GT in 2000 to 8.2 GT in 2018. Despite the gradual improvement in global energy efficiency, the popularization of electrification, and the more scientific use of biofuels, $CO_2$ emissions still maintained an annual growth of 1.9% by 2020. Among them, road vehicles—cars, trucks, buses, two-wheeled and three-wheeled vehicles—account for three-quarters of the $CO_2$ produced by the transport sector. As a solution to reduce energy consumption and environmental impacts in the transportation sector, NEVs (including EVs, PHEVs, HEVs, FCEVs, EREVs, etc.) are gradually advancing and replacing ICEVs. By the end of 2020, the number of NEVs on the road around the world reached 10 million and had experienced rapid growth for 10 years. From the perspective of the government [1] and environmental protection agencies [2], NEVs are ideal vehicles approaching zero pollution and zero emissions and are a powerful advantage for achieving environmental, social,

and health goals [3]. Their zero exhaust emissions are very suitable for alleviating air pollution [4] and the dependence on fossil fuels [5].

However, it seems that NEVs do not conform to the ideals of zero-pollution and zero-emission vehicles vaunted in government reports [6]. The driving emissions of NEVs transfer the burden of driving emissions to the power plant. Therefore, the power generation configuration and power generation emissions of a country or region greatly affect the environmental improvement efficiency of NEVs in the relevant country or region [7–10]. Moreover, NEVs will still increase environmental pollution, such as by particulate matter formation (PMF) [11,12], human toxicity (HT) [13,14], and terrestrial acidification (TA) [12,15]. As the first batch of NEV batteries reaches the scrap standard threshold, the recycling and reuse of old batteries and the manufacture of new batteries will cause further pollution [16,17]. Therefore, an evaluation of whether NEVs are a strong choice to achieve environmental, social and health goals should look at not only the driving stage of the vehicle but also the upstream manufacturing stage [18], the energy supply stage [19], and the downstream recycling and reuse stage [20].

As a method recommended by ISO14040 and ISO14044 in 2006, life cycle analysis (LCA) aims to find opportunities for a product or service to improve the environmental performance of that product at all stages of its life cycle, providing a choice of environment-related indicators, to provide a reference for technical and marketing environmental efficiency [12,21–23]. Many researchers have tried to use LCA to evaluate the environmental efficiency of NEVs, to prove the difference in environmental efficiency between NEVs and ICEVs from a more comprehensive perspective [24–26]. LCA not only focuses on the evaluation of environmental efficiency but also on life-cycle cost (LCC) [27], life-cycle inventory (LCI) [28], and social life-cycle analysis (S-LCA) [29], which expands the evaluation scope of LCA.

Currently, the use of LCA studies regarding NEV environmental efficiency results is not consistent. We have selected a few papers from between 2019 and 2021 to show the results given for $CO_2$ emission and the standpoint about whether or not NEVs are cleaner than traditional ICEVs, as shown in Table 1. Most researchers have affirmed the environmental benefits of NEVs, especially in reducing the greenhouse effect [7], but in Poland and the Czech Republic, the widespread use of NEVs will cause current and future GWP and fossil fuel consumption to be lower than those of ICEVs but will increase acidification, eutrophication, HT and particulate matter levels [30]. In Brazil, the direct use of thermal power generation to provide energy for NEVs will reduce the environmental benefits brought by NEVs [31]. When taking into account battery aging and the replacement of NEVs, the $CO_2$ emissions of NEVs will rise significantly, to be even higher than those of mature ICEVs [32].

**Table 1.** Studies taking the standpoint of whether NEVs are cleaner or not.

| Year | Author | $CO_2$ of NEVs (gCO$_2$-eq/km) | $CO_2$ of ICEVs (gCO$_2$-eq/km) | Standpoint |
|------|--------|--------------------------------|---------------------------------|------------|
| 2021 | Nimesh V [24] | 187 | 215 | YES |
| 2021 | Petrauskiene K [33] | 212 | 159 | NO |
| 2020 | Koroma MS [18] | 170 | 213 | YES |
| 2020 | Liu YT [34] | 244 | 92 | NO |
| 2020 | Qiao QY [35] | 253 | 340 | YES |
| 2020 | Petrauskiene K [36] | 142 | 104 | NO |
| 2019 | Kim S [11] | 100 | 170 | YES |
| 2019 | Cusenza MA [37] | 240 | 180 | NO |
| 2019 | Shen W [38] | 163 | 199 | YES |
| 2019 | Almeida A [39] | 141 | 193 | YES |

Other authors have conducted a detailed literature review of the published research on using LCA to evaluate the environmental efficiency of NEVs from various angles and compared the research findings on the environmental efficiency of NEVs and ICEVs [40].

Studies on the reliability of NEVs [6], NEVs and smart grids [41], the manufacturing and recycling of lithium batteries [42,43], and the LCC of NEVs [44] found that different manufacturing data selection, lack of timeliness, targets, and modeling selection all affect the authenticity of the results [6]. Presently, review studies tend to study a certain stage, focusing on the energy stage, manufacturing stage, or recycling stage of a product. In the whole cycle, this encompasses which stages are considered the key points of measurement, how to choose indices and methods, what trends the results show, and which evaluation directions are to be fully and clearly analyzed. To answer these questions, the current study reviewed 282 papers from the core set of the Web of Science (WOS) that used LCA to evaluate the environmental efficiency of NEVs. In this paper, we identify the commonly used indicators and models, construct evaluation formulations, find the weak points of previous research and predict the direction of future research.

## 2. Methodology

### 2.1. Data Selection

This article uses data collected from the WOS. The WOS is the most authoritative citation database in the world and is also the most reliable database in terms of bibliometric research [45]. The time span of the search was from the beginning of the database, November 1996, to the end of May 2021.

The scope of the literature search was first determined; except for the search keyword of LCA, "new energy vehicles" and "energy conservation and environmental protection" are all search terms, and among them, "new energy vehicles," including EV, PHEV, FCEV, HEV, and EREV. Considering that the research using LCA to analyze electric vehicle environmental efficiency is extensive, the terms "new energy vehicles" and "electric vehicles" were searched together. Similarly, a search was also initiated for "energy conservation and environmental protection", among which "energy conservation and environmental protection" includes the terms "environmental", "sustainable" and "ecological" [46]. In total, 534 studies were ultimately collected; we excluded studies that did not conform to the research content or did not emphasize the energy-saving or environmental protection characteristics of NEVs. A total of 282 articles were screened from 1996 to May 2021; the details about the titles and DOIs of 282 articles are shown in Supplementary Materials.

### 2.2. Annual Publications

Figure 1 shows the number of papers published from November 1996 to the end of May 2021. The papers published pre-2000 are indicated as "Before 2000". The first study on using an LCA evaluation of new energy automotive environmental performance can be traced back to 1996, this article, written according to ISO14040 and originally on LCA, assessed the four types of battery production, supply and recycling energy, and material flow analysis, and considered the GWP, ozone depletion (OD), terrestrial acidification (TA), eutrophication, human and ecological toxicity. The environmental efficiency of FCEVs was evaluated from the perspective of the life cycle [47], written when the concept of LCA was not yet mature. In 2006, ISO14040 and ISO14044 defined LCA, LCI, and life-cycle impact analysis (LCIA) and replaced ISO14040 in 1997. The combination of LCA and NEVs to evaluate environmental efficiency has gradually attracted attention.

### 2.3. Country Publications

Figure 2 shows the number of journals published by each nationality. American researchers and Chinese researchers have provided roughly the same research input in this direction. Although the United States began to evaluate the environmental efficiency of NEVs and diesel vehicles from the perspective of the full life cycle in 2001, the number of newly registered NEVs in the United States in 2020 was 295,000, far less than China's 1.159 million. This figure shows that Chinese researchers have gradually caught up with American researchers. Compared with other European countries, Germany not only registered 3.94 million NEVs in 2020, ranking first in Europe, the number of publications is

also ranked first in Europe. In addition, it was found that India and Malaysia also appear in the table, which was related to Indian researchers [48] and Malaysian researchers [49] studying the environmental efficiency of NEVs that were independently developed and produced by their own countries.

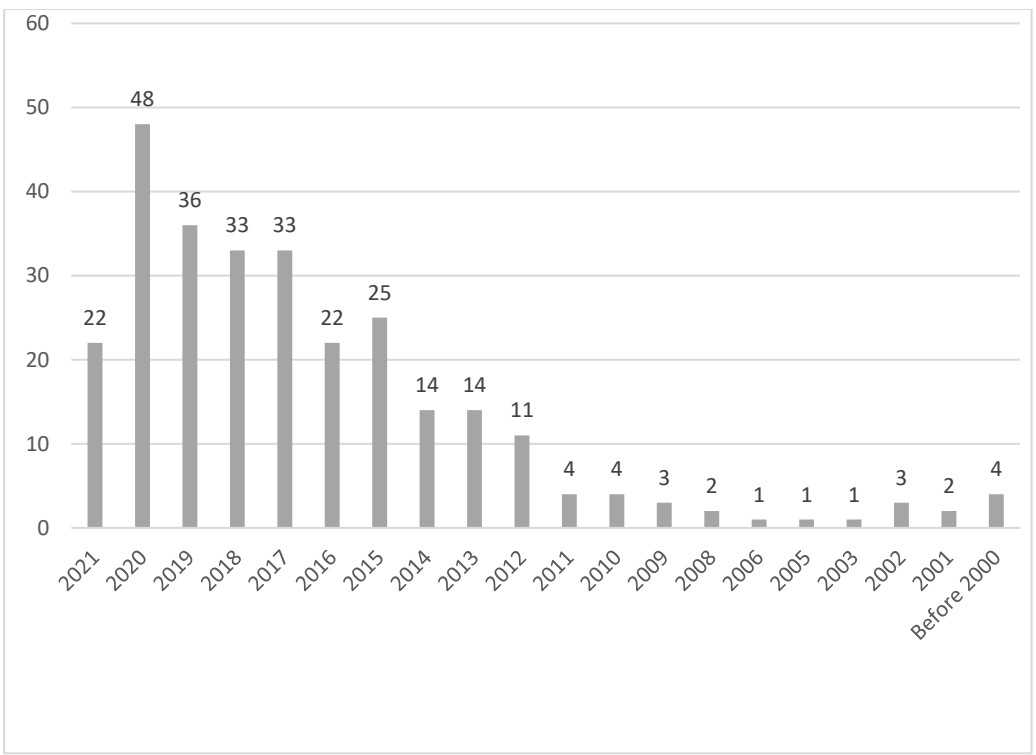

**Figure 1.** Yearly distribution of papers.

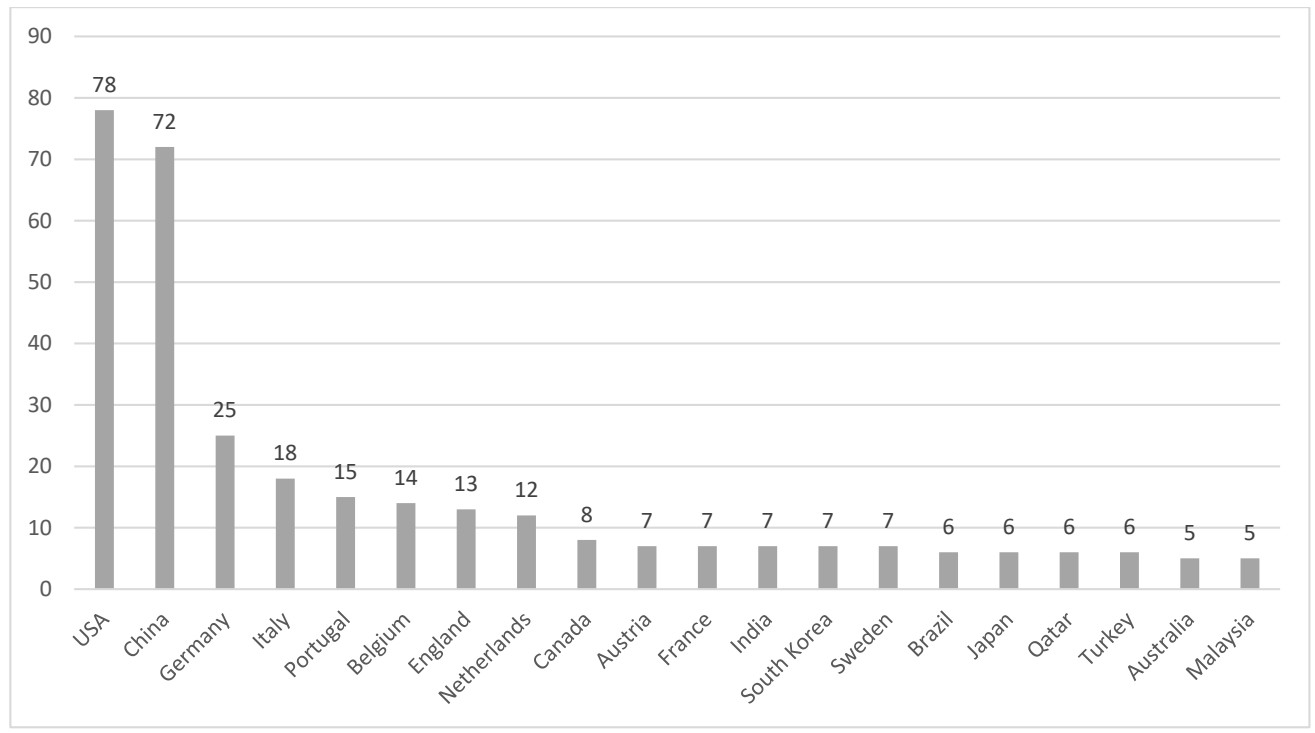

**Figure 2.** Distribution of published papers by country/region (top 20).

### 2.4. Vehicle Discussion

Figure 3 shows the word frequency statistics of each paper discussing NEVs and ICEVs. Among them, NEV papers include EV, PHEV, FCEV, EREV, and HEV, ICEV papers include GICEV, diesel (diesel internal combustion engine vehicle) [50], LPG (liquefied natural gas vehicle) [51], and CNG (compressed natural gas vehicle) [52]. The use of LCA to evaluate the environmental efficiency of ICEVs is also very popular because ICEVs are usually used as a control group to demonstrate that the implementation of EVs can improve environmental efficiency. PHEV and HEV appear almost the same number of times because they are usually discussed together [8,53], while FCEV papers mainly focus on biofuel and HFCEVs. EREVs, which are vehicles with a fuel type between PHEVs and EVs, offer significant advantages in terms of saving mineral resources and fossil energy, with the mineral resource consumption of EREVs being 14.68% lower than that of HEVs and the fossil energy consumption being 34.72% lower than that of ICEVs [34].

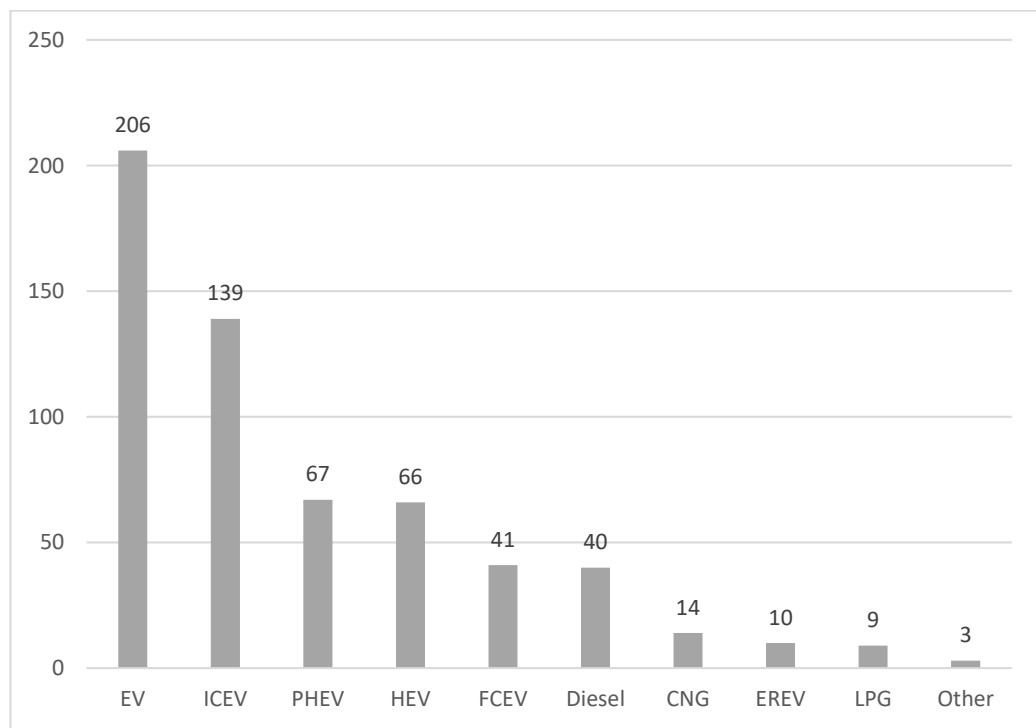

**Figure 3.** The number of vehicle types discussed.

## 3. Environmental Efficiency Evaluation Process of NEVs under LCA

### 3.1. LCA Evaluation Method Division and Selection

To evaluate the environmental efficiency of NEVs from the perspective of the life cycle, the evaluation method should be determined first. In addition to LCA, LCI and LCIA are also mentioned in ISO14040 and ISO14044. LCI measures the life cycle efficiency inventory of products or services from the perspective of material flow, while LCIA uses LCI results to evaluate the environmental impact of potential products or services. LCC follows the cycle division concept of LCA and mainly measures the cost of products or services, while SLCA is similar in that it measures social benefits. There are even life cycle sustainability assessments (LCSA) that combine LCA, LCC, and SLCA [3,54] and macro approaches that use an extended input–output life cycle evaluation of the interaction of multiple industries (EIO-LCA) [55]. Table 2 presents the statistics of the LCA and LCA expansion methods. Statistics for all entries related to LCA in the study were created.

**Table 2.** Choices of evaluation methods in the various studies.

| Rank | Method Selection | Numbers |
| --- | --- | --- |
| 1 | LCA | 233 |
| 2 | LCI | 79 |
| 3 | LCC | 63 |
| 4 | LCIA | 35 |
| 5 | SLCA | 6 |
| 6 | ALCA | 2 |
| 7 | CLCA | 1 |
| 8 | PLCA | 1 |
| 9 | EIO-LCA | 1 |

As seen from Table 2, although all studies considered LCA, not all studies used the actual term LCA, instead using, for example, well to tank (WTT), tank to wheel (TTW), or well to wheel (WTW) [56], or they only considered the production [57] or recovery phases [58]. These studies use the life cycle concept rather than the whole LCA. As an analysis model of material flow, LCI subdivides products into their raw materials. Limited by LCI, LCIA can only deal with those objects selected by LCI and cannot conduct environmental evaluation from a more holistic perspective, as is possible with LCA [59]. When input and output values pay more attention to causality and process connection, attribution life cycle analysis (ALCA) [55,60,61], indirect life cycle analysis (CLCA) [61], and process life cycle analysis (PLCA) [55] gradually appear in the field of vision of researchers, but they are all in the research concept construction stage.

LCC follows the life cycle concept of LCA and conducts statistics assessment and analysis on the cost instead of the environment. The most representative cost is the private life cycle cost, which is divided into tangible costs and intangible costs. Tangible costs include the purchase cost (PC), operating cost (OC), and recovery/resale value (RV) [62–64], wherein purchase cost is also known as the total cost of ownership (TOC), including purchase price, sales tax, ownership fee and any subsidy or tax refund [65–67]. Intangible costs include the purchase restriction intangible cost (PRIC) and driving restriction intangible cost (DRIC), such as the license plate lottery, driving limit, and consumer evaluation [68]. In addition, the battery replacement cost [69], safety cost [70], enterprise average fuel economy (CAFE) [71], hydrogen fuel cost [66], time cost and learning cost are all details that should be considered [63,66,72]. In some studies, environmental impact is included in the LCC evaluation, and the impact of environmental issues on cost is considered in the form of $CO_2$ cost [70,73–75], carbon price [76], and carbon index [77]. Cost measurements of other forms of environmental pollution are also involved, but they all focus on air pollutants [5].

The SLCA social life cycle is the process of using a social standpoint to measure the social benefits of NEVs [26]. In the current stage, NEVs' and ICEVs' social influences are different; ICEVs no longer receive more attention from the government [26] but involve less toxicity to humans [54]. NEVs have created more employment [54] but have dropped in consumer preference due to immature technology and range anxiety [26].

### 3.2. Stage Division

The basic automobile LCA process model constructed by Nordelof [6,78] was followed as shown in Figure 4. Table 3 shows the number of times that each stage is included in the research boundary. Among them, the WTT energy phase includes energy resource extraction and collection, carrier production and energy transportation, and energy refinement and distribution, while the TTW energy phase includes driving route and energy conservation. Researchers pay less attention to the maintenance and parts replacement stages, due to the difficulty of obtaining data and the lack of a unified scale.

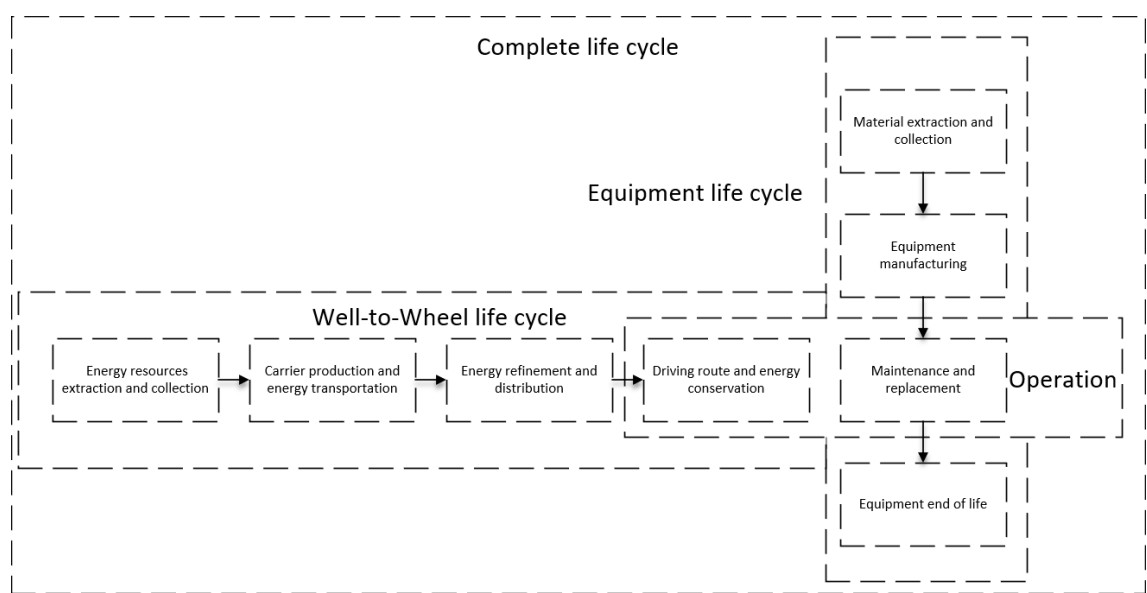

**Figure 4.** Basic vehicle life-cycle analysis model.

**Table 3.** Stage selection frequency.

| Rank | Stage Division | Numbers |
|------|----------------|---------|
| 1 | Driving route and energy conservation | 206 |
| 2 | Equipment manufacture | 154 |
| 3 | Energy refinement and distribution | 135 |
| 4 | Equipment end of life | 92 |
| 5 | Material extraction and collection | 88 |
| 6 | Energy resources: extraction and collection | 86 |
| 7 | Carrier production and energy transportation | 56 |
| 8 | Maintenance and replacement | 52 |

### 3.2.1. Energy Stages

Energy Resource Extraction and Collection

The extraction of oil, coal, and natural gas and the production of hydrogen, biogas, and ethanol are all counted in the energy extraction and capture phase, where the key measure of efficiency is energy.

The unit MJ/MJ was used in this study; that is, the energy of standard coal is used to measure the value of mining or producing another kind of energy [79,80], such as biogas [74] and landfill gas [81], hydrogen [82,83] and ethanol [84]. In the case of biogas and landfill gas, this is suitable for combining with natural gas. When the production is located near the natural gas network, biogas can be efficiently transported through the natural gas network and used for natural gas-powered vehicles [74,81] or power and heat production in thermal power plants [74] to achieve the highest environmental efficiency. Changes in production processes and the energy sources required for production lead to differences in different energy conversion (such as "sunlight to oil" [85]). When the proportion of thermal power is reduced to less than 40%, the environmental benefit of hydrogen production from the electrolysis of water is expected to rise to be equal to that from the electrolysis of steam methane [82]. The environmental efficiency of ethanol production is also affected by its energy sources. The energy consumption of ethanol production is 0.51–0.84 MJ/MJ, and the $CO_2$ emissions are 39.44–49.97 g $CO_2$-eq/MJ [80]. These two values are sensitive to the thermal power ratio.

Carrier Production and Energy Transportation

Energy transport, in which mined energy is transported from upstream to downstream via pipelines, vehicles, or power lines, considers the uncertainties and losses encountered during transport. At this stage, environmental efficiency is measured in losses.

The greenhouse effect of NEVs, especially EVs, depends on both power consumption and power grid transmission [86]. The loss of power grid transmission varies according to the infrastructure construction of countries, for example, 6% in China [35] and 6.5% in the United States [87]. Pushing NEVs triggers the "neighbor effect"; imported parts can easily reduce the pollution of the local environment and avoid energy export, and other regions must accommodate the burden of higher energy consumption, since reducing environmental pollution in production and acquisition work is more difficult [88]. The $CO_2$ emission of natural gas used in local and produced in another place is 40% higher than that of used in local with local production, which is influenced by transportation loss [19,89]. Whether it is gaseous or liquid transportation, using ordinary compressed gas trailers, liquid hydrogen pipelines, or liquid hydrogen vehicles [82], the $CO_2$ emissions of HFCEVs in the WTW stage are at least 15% lower than those of ICEVs [90]. Updating carbon fiber hydrogen storage tanks to transport hydrogen can further improve the environmental efficiency of hydrogen fuel cell vehicles [91].

Energy Refinement and Distribution

There are many sources of energy, and each energy source has a different means of improving efficiency and spatial-temporal configuration. The environmental efficiency assessment of this stage was further refined. At this stage, environmental efficiency was measured by efficiency versus the ratio. Efficiency includes energy efficiency [80,92,93] and charging efficiency [94,95]. The ratio usually refers to the power generation allocation ratio, which means the ratio of each power generation mode to the total power generation of a region.

Whether measuring fossil energy or renewable energy, the energy conversion efficiency is an important index by which to measure energy and environmental efficiency [79]. Different energy sources have different energy conversion rates [80,92]. Charging efficiency includes the charging efficiency of the charging pile and also battery discharge efficiency. Due to the difficulty of obtaining data, either the GREET database [96] was used or charging efficiency was assumed to be consistent with discharge efficiency [94].

The power structure determines the energy distribution level of a region and can intuitively reveal the degree of energy cleanliness of a region. However, this ratio can change according to the region [97], season [98], time period [94], and other factors. Denmark, Germany, and other European countries have an environmental efficiency advantage because the wind power, solar power, nuclear power, and tidal power of clean energy are higher [97]. Because solar radiation is highly influenced by day length and season, and wind power is influenced by the monsoon [7], the share of solar energy and wind energy can drop from 33.5% in summer to 5.1% in winter [95], so thermal power stations are needed to compensate for the loss of energy. Therefore, environmental efficiency decreases [94]. However, the power structure fluctuates continuously [7] and is constantly optimized by policies [79], so it is unreasonable to use static and post hoc data to measure prior LCA research, which requires the use of big data and the construction of a more real-time and authoritative database [22]. In the face of an energy crisis, good environmental efficiency performance may be lost as a shortage of dependent fuel resources forces changes in the power mix [99].

### 3.2.2. Production Stage
Material Extraction and Collection

Before the production of auto parts, mining and extraction of the most basic metal resources, such as iron ore and copper ore, nonmetallic resources, such as carbon fiber, graphite, and lightweight materials, and processing them into semi-finished products are

the first steps in the production stage. At this stage, environmental efficiency is measured by material flow.

LCI is generally used to analyze the environmental efficiency of NEVs in raw-material mining and extraction. The raw materials required for assembly into NEVs are divided into metal [100] and nonmetal [53,101], and the environmental efficiency of each step of raw materials from mining to processing of semi-finished products were studied in terms of material flow [102]. Among the material flows discussed, copper [103,104], iron [18], and lithium [105] are the most discussed material flows, and different mineral sources [106] can lead to different environmental efficiencies. The material flow environmental efficiency of each material is different. The environmental impact of lithium carbonate material flow occupies a very small proportion of the production stage of pure electric vehicles [106], but the material flow of cobalt can greatly affect the environmental efficiency of production [107]. Material flow analysis can find the weak points of material flow in the production process [53,108]. Considering the hidden recovery potential of metal material flow, the stability risk of the supply chain can be greatly reduced, particularly for lithium iron phosphate batteries (LiFePO$_4$) and lithium nickel cobalt manganese acid (NMC) [109]. However, there is also a material flow, namely, of permanent magnet rare earth, with low consumption but huge stability and environmental and social risks because of its specific suppliers [100,110].

Equipment Manufacturing

After raw material mining and extraction, equipment manufacturing determines the ratio of raw materials required by different parts, on the one hand, considering the operational efficiency and environmental efficiency of different equipment on the other hand. At this stage, environmental efficiency is measured by efficiency and ratio, and parts can be divided into auto parts and NEV batteries.

Automobile components can be divided into five groups, including the powertrain/engine, transmission system, glider, motor/generator, and battery [111], and component efficiency is mainly composed of the generator and frequency converter, gear transmission, wheel and axle, and brakes [93,112]. Energy consumption depends on the efficiency of the power system. Improving efficiency should be one of the keys to increasing environmental efficiency [112]. Efficiency can be improved by using lightweight materials (such as carbon fiber) [101,113], improving component efficiency and heat dissipation rate [103], changing component chemical composition [112,114], building dynamic models to study efficiency breakthroughs [115], or adding energy components [116]. However, emphasis should be placed on improving the efficiency of the motor/generator [113].

For NEVs, among the environmental benefits of all parts production, the battery system has the greatest impact on the environment. The whole battery system includes the cathode, anode, electrolyte, separator, binder, battery management system, cooling system, and packaging [108]. The copper used in the anode and gold used in the integrated circuit account for 40% and 26% of the environmental efficiency, respectively [103]. To improve the environmental efficiency of battery production, the core strategy is to improve the heat dissipation rate [103]. Using ultracapacitors [57,117], optimizing the number of battery packs [118] or material ratio [25,39], using new materials for batteries [39,119], and optimizing the battery management system [103] can improve the environmental efficiency of the battery system to a certain extent.

### 3.2.3. Operation Stage
### Driving Routes and Energy Conservation

The whole stage of the NEV from delivery to the hands of consumers, to driving and scrapping, is the content that needs to be considered in the driving process and in energy conversion. This stage not only includes the performance of the vehicle in operation but also consumers' driving and charging preferences, weather and road conditions, seasonality and time differences of electricity consumption, and different control policies all affect the

environmental efficiency of NEVs in this stage. As the most extensively studied stage, environmental efficiency is measured by mileage change and environmental uncertainty.

Theoretically, the cubic power of vehicle driving power and speed is proportional to the driving resistance [120,121], reducing the front projection surface [93], reducing brake loss [99], or increasing vehicle specific power [113] can optimize the energy demand, thus affecting the vehicle mileage. The driving style is different for NEVs and ICEVs [95]. A non-aggressive driving style is more suitable for NEVs because NEVs need to store braking energy [93,99,122], while a more aggressive driving style will increase energy consumption by 47% [94]. Compared with ICEVs, EVs are more suitable for urban environments [123] or driving environments with short distance distributions [67], while PHEVs [102] or EVs loaded with larger batteries [113] are more suitable for driving environments with large slopes [120]. However, the user driving mode of ICEVs is not subject to the change of mileage limit [102] and single driving distance [123].

In a day, the electricity price is different at different times [124]. In Germany, the electricity price at peak periods (2 p.m. to 8 p.m.) is 2.5 times that in the middle peak period (7 a.m. to 2 p.m. and 8 p.m. to 10 p.m.) and 3.3 times that in the nonpeak period (10 p.m. to 7 a.m. the next day) [125]. Therefore, the charging preferences of car owners will change according to time and space factors. Charging transaction data can comprehensively reflect the charging preferences of car owners [126], identify areas lacking charging infrastructure, elucidate the utilization and modes of charging behavior, and distinguish charging preferences at different times and in different regions [99]. Charging at night emits more $CO_2$ than charging during the day [99]. Charging during off-peak hours can reduce $CO_2$, PM2.5, $NO_x$ and $SO_2$ emissions by 12%, 15%, 13% and 12%, respectively, during the operation phase [127], and different temperatures [128] can also affect each charge preference.

Environmental uncertainty makes the emission change of NEVs more drastic, including technological change [129], policy change [68] and emergencies [130]. Internet plus can realize the interaction between electric vehicles and the power grid, with the help of smart grid technology and internet intelligent control devices, so that this uncertainty is gradually controllable [129]. When the current subsidy policy has little effect due to emergency situations such as the COVID-19 pandemic, technology-centered incentive and subsidy policies can achieve environmental and income balance at the fastest speed [130]. However, technological innovation that improves efficiency often fails to give full play to its potential when considering the environmental rebound effect [131].

Maintenance and Replacement

To ensure the stable operation of NEVs, it is necessary to maintain and replace parts in a timely fashion, which will reduce the environmental efficiency of NEVs. At this stage, environmental efficiency is measured by the replacement cycle. The aging of components is a continuous process [132,133] but it is often ignored for the convenience of calculation or for lack of specific information [134]. More studies assume that the working efficiency of components remains unchanged before replacement [28]. In general, 40% of maintenance costs are spent on parts procurement, and the rest are concentrated on manual labor [135]. Maintenance parts, except tires, power systems, coolants, brake fluids and windshields [136], as well as batteries are the focus of maintenance and the parts replacement of NEVs [128,134].

3.2.4. Equipment Ends of Life

When NEVs reach the end of the equipment's life, NEVs themselves and their replacement parts need to be destroyed, recycled, and reused, among which the recycling and reuse of waste batteries have always been the focus. At this stage, environmental efficiency is measured by efficiency, whether it is material recovery or reproduction and utilization. Among them, material recovery, reproduction, and the reuse of NEV batteries are relatively cutting-edge research topics [137,138].

The material recycling of batteries reduces battery pollution, reduces battery costs, and reduces dependence on specific suppliers, especially to compensate for the growing gap in metal inventory (mainly nickel, cobalt, manganese, lithium, copper, aluminum, and iron) [109]. However, it is difficult to find reliable recyclers, presenting high recovery costs and low technical environmental efficiency at present [16]. Average battery material recycling can reduce 22% of the production environment, including NMC recovery efficiency, which is highest because the copper and cobalt-nickel sulfate and sulfuric acid secondary raw materials have high availability [139].

Battery reuse is still an untapped market [20]. A second-life battery is mainly used in renewable power generation technology and is reused in residential buildings, improving the high variability of renewable energy generation and matching building electricity demand [58,140]. Compared with the direct use of new cells, second-life cells reduce power costs by 12–57% and $CO_2$ by 7–31%, which can be reduced even more in industrial applications [141], or the construction of large energy-storage devices [58]. Additionally, using a second-life battery for rooftop photovoltaic energy storage has more valuable environmental benefits [20], all of which must take into account second-life safety features [142].

### 3.3. Types of Research Objects and the Selection Trend of Performance Indices

Table 4 shows the database and analysis software used in the selection of research object types and indicators. GREET was specifically used to evaluate automobile LCA and mainly evaluates the WTW process of NEVs [8]. Simapro and Gabi are powerful tools for measuring LCA, LCI, and LCIA. It was also used elsewhere in combination with the Ecoinvent database for the statistics of various links of LCI and LCIA [14,22]. In addition to these databases and software, researchers also use more local, diverse or professional databases and software, such as TLCAM [52,143] and CALCD [144], which are customized to evaluate the environmental efficiency of NEVs in the Chinese environment. In the Japanese environment, MiLCA software and the environmental IDEA database [138] are used. Other software that is used includes the water efficiency measure NREL [145], the LCC measure AFLEET [5], the traffic conditions measure EVRO [5], and the vehicle driving simulator ADVISOR [113]. The rest of the software can be divided into two categories. One category is similar to Simapro or Gabi LCA evaluation software, for example, OPENLCA [91], CMLCA [146], COPERT5 [118], Umberto [102], ELCD [22], MRIO [3] and CNMLCA [147]. The other category is a dataset similar to Ecoinvent, including IEA [22], JEC [83], EXIOBASE, E3OIT, and WIOD [131].

#### Selection Trend of Vehicle Types and Performance Indices

Statistics were calculated on the automobile brands used in the study (Table 5) and the most popular cars (Table 6). It was found that Toyota was selected for the most studies. Among them, the research cases of Honda focused on the Toyota Prius HEV and PHEV, the Toyota Corolla for ICEV, and the Toyota Mirai for HFCEV. Compared with other Japanese automobile makers, Toyota has a more diversified selection than Nissan. The Nissan Leaf is the most representative compact electric car in the study. This is because the production data of the Nissan Leaf on Ecoinvent are relatively comprehensive [33,99,130].

The emerging NEV manufacturers Tesla and BYD are also hot research objects. In particular, the Tesla Model S [130,148] and studies on electric vehicle enterprises specializing in the local market, such as BAIC [55] and Geely [88], have gradually received more attention.

The performance of each vehicle is different, and the selected research indicators should of course be considered. Detailed data for all indices are given in Supplementary Materials.

For the convenience of image representation, EV is classified as pure electric vehicles, PHEV, HEV, EREV, and FCEV are classified as hybrid electric vehicles (hereafter hybrid), ICEV with a gasoline engine is classified as either GICEV or diesel ICEV, and LPG and CNG are classified as other internal combustion engine vehicles (OICEV); this setting is followed in the rest image. The total mileage statistics of vehicles are shown in Figure 5. All studies unify the maximum mileage of vehicles involved in their respective studies, and

the maximum mileage gradually settles upon 150,000 km. However, the previous mileage was overly pessimistic, such as 200 km [21], or overly optimistic mileage, for example, at 500,000 km [15,62] the maximum value from 2014 to 2021 is concentrated at approximately 240,000 km. The unit used in these studies is the mile, but the threshold of 150,000 km is still used [77,121]. The single farthest driving distance of vehicles is shown in Figure 6. The shortest driving distance is focused on EV, and the average farthest driving distance increases steadily from 225 km in 2011 to 310 km in 2021. However, the driving distance of a hybrid vehicle is generally set at longer than that of an EV, especially 486 km in 2021. The driving range of PHEVs is divided into pure oil mileage and electric mileage, but more attention is given to electric mileage [136,149].

**Table 4.** Database and evaluation software selection.

| Rank | Database/Software | Numbers |
|:---:|:---:|:---:|
| 1 | Greet | 62 |
| 2 | Ecoinvent | 52 |
| 3 | Simapro | 29 |
| 4 | Gabi | 22 |
| 5 | TLCAM | 6 |
| 6 | OPENLCA | 5 |
| 7 | AFLEET | 3 |
| 8 | Umberto | 2 |
| 9 | MRIO | 2 |
| 10 | NREL | 2 |
| 11 | EXIOBASE | 2 |
| 12 | E3OIT | 2 |
| 13 | CMLCA | 1 |
| 14 | CALCD | 1 |
| 15 | MiLCA | 1 |
| 16 | COPERT5 | 1 |
| 17 | ELCD | 1 |
| 18 | IEA | 1 |
| 19 | JEC | 1 |
| 20 | CNMLCA | 1 |
| 21 | ADVISOR | 1 |
| 22 | WIOD | 1 |
| 23 | IDEA | 1 |
| 24 | EVRO | 1 |
| 25 | No instructions | 100 |

**Table 5.** Vehicle brand selection (top 20).

| Rank | Vehicle Brand | Numbers |
|:---:|:---:|:---:|
| 1 | Toyota | 53 |
| 2 | Ford | 40 |
| 3 | Nissan | 40 |
| 4 | Volkswagen | 37 |
| 5 | Tesla | 24 |
| 6 | BYD | 23 |
| 7 | Chevrolet | 18 |
| 8 | Honda | 18 |
| 9 | BMW | 17 |
| 10 | BAIC | 16 |
| 11 | Hyundai | 16 |
| 12 | Kia | 10 |
| 13 | Chery | 9 |
| 14 | Mercedes-Benz | 9 |
| 15 | Mitsubishi | 8 |
| 16 | Fiat | 7 |
| 17 | Renault | 7 |
| 18 | Smart | 7 |
| 19 | Volvo | 7 |
| 20 | Geely | 6 |
|  | Total | 372 |

**Table 6.** Vehicle model selection (top 10).

| Rank | Vehicle Model | Numbers | Power Type |
|------|---------------|---------|------------|
| 1 | Nissan Leaf | 40 | EV |
| 2 | Toyota Prius | 14 | HEV |
| 3 | Tesla Model S | 12 | EV |
| 4 | Volkswagen Golf | 9 | EV |
| 5 | Toyota Prius | 9 | PHEV |
| 6 | Toyota Corolla | 8 | ICEV |
| 7 | Toyota Mirai | 7 | FCEV |
| 8 | Volkswagen Golf | 6 | ICEV |
| 9 | BMW i3 EV | 6 | EV |
| 10 | BYD e6 | 5 | EV |

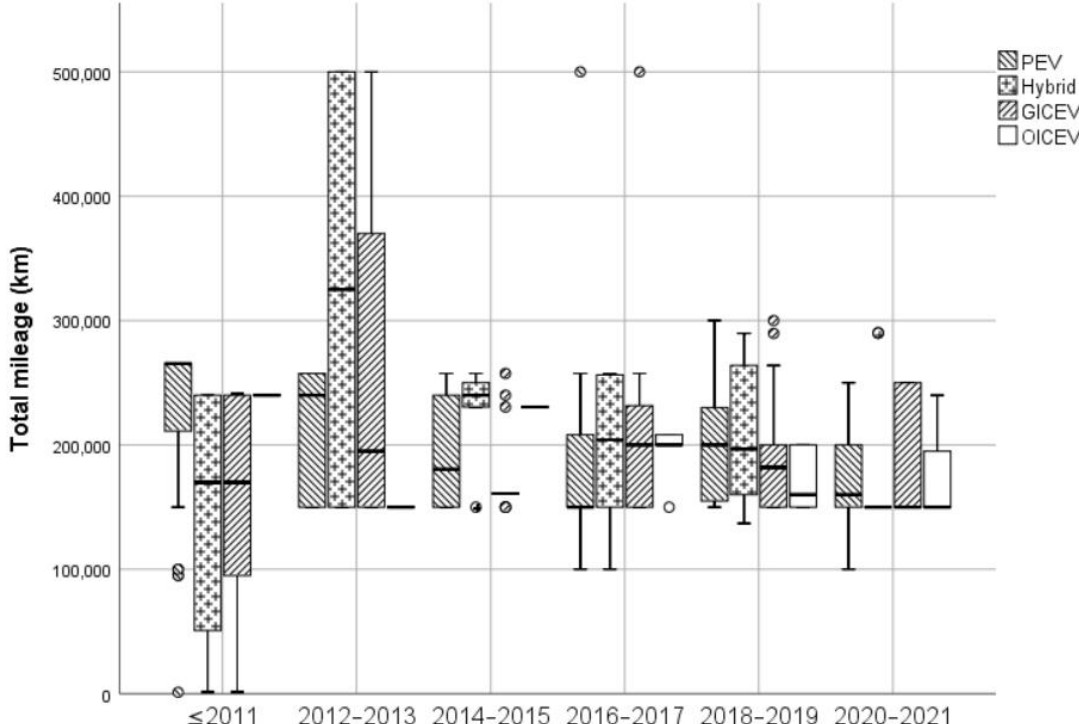

**Figure 5.** Total mileage selection.

It is more reasonable to use energy consumption directly (Figure 7) to measure energy consumption between different types of vehicles, such as the energy footprint [150]. However, such data need to consider the power mix, the calorific value of different fuels [79], and transmission loss [96], which is not intuitive and is difficult to obtain. Therefore, more researchers choose 100 km fuel consumption (Figure 8), 100 km power consumption (Figure 9), battery energy storage (Figure 10), and 100 km hydrogen consumption (Figure 11) as the bases for the evaluation of basic energy consumption. It was found that the fuel consumption per 100 km of both hybrid and GICEV is decreasing. The average fuel consumption of gasoline internal combustion engine [151] vehicles changed from 6.39 L/100 km in 2016 to 5.93 L/100 km in 2021. However, the study on SUVs declares a fuel consumption (14.5 L/100 km) significantly different from the average. Due to the small sample data on SUVs, the result implies distortion [3]. With the increase in research data and technological progress, the 100 km power consumption and battery energy storage of pure electric vehicles have experienced a process of first rising and then falling. The average 100 km power consumption increased from 12.6 kWh/100 km in 2011 to 18.15 kWh/100 km in 2016 and then to 16.55 kWh/100 km in 2021. The average battery storage value increased from 31.1 kWh in 2011 to 28.59 kWh in 2016 and 43.46 kWh in 2021. The statistical sample size of 100 km power consumption and battery energy storage of hybrid power is small, the trend is not stable, and the extreme value

is large [3,37] because the battery of a hybrid is generally small and only plays an auxiliary power role. The paper on HFCEV energy consumption can be traced back to Georgakellos' research, but the value given is too large (12.8 L/100 km) due to immature technology [21]. After the technology stabilized, it stabilized at 0.85 L/100 km in 2021 [34].

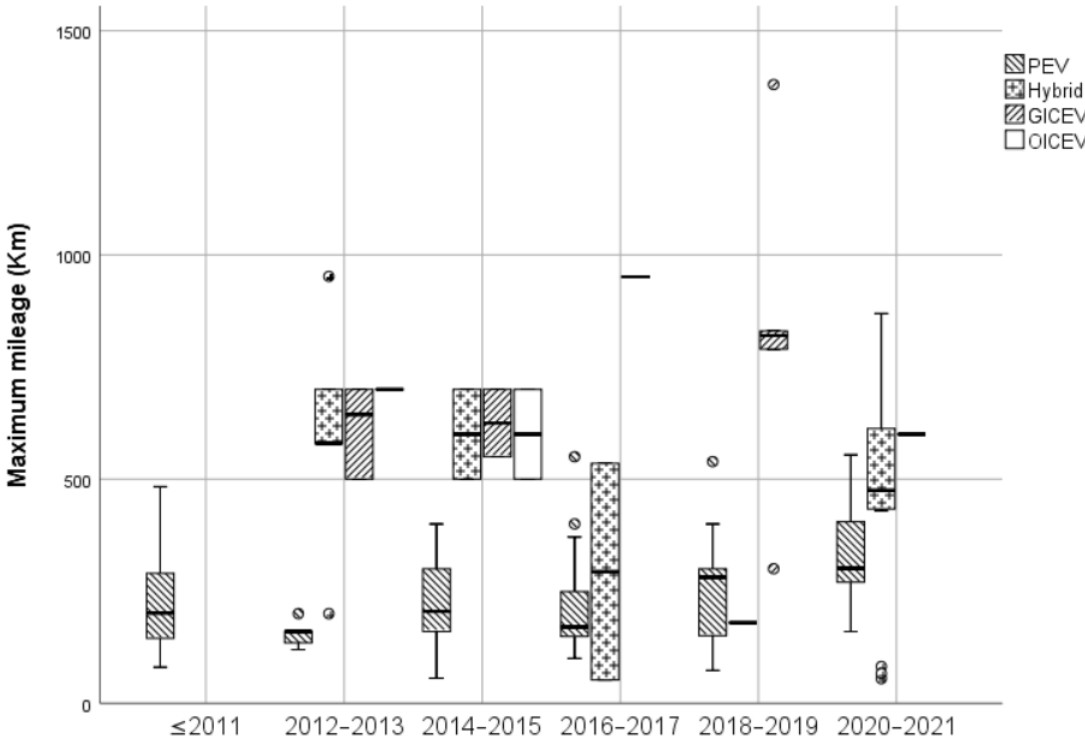

**Figure 6.** The maximum mileage of the vehicle.

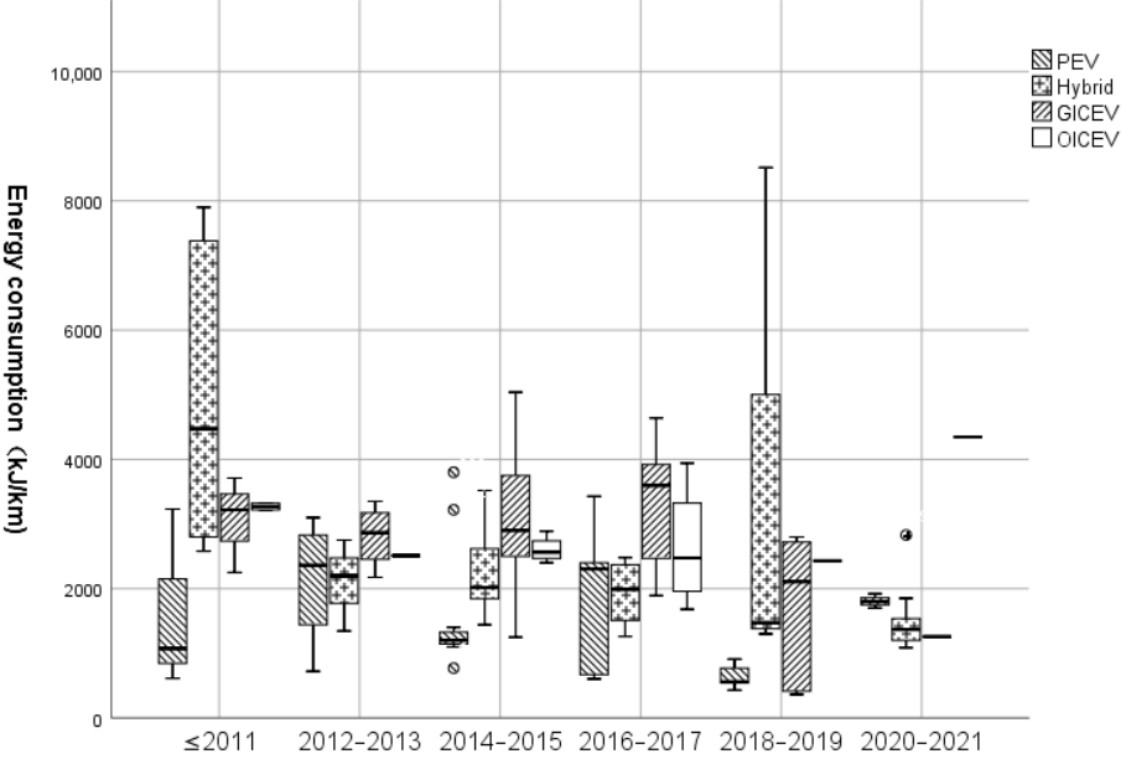

**Figure 7.** Energy consumption per kilometer.

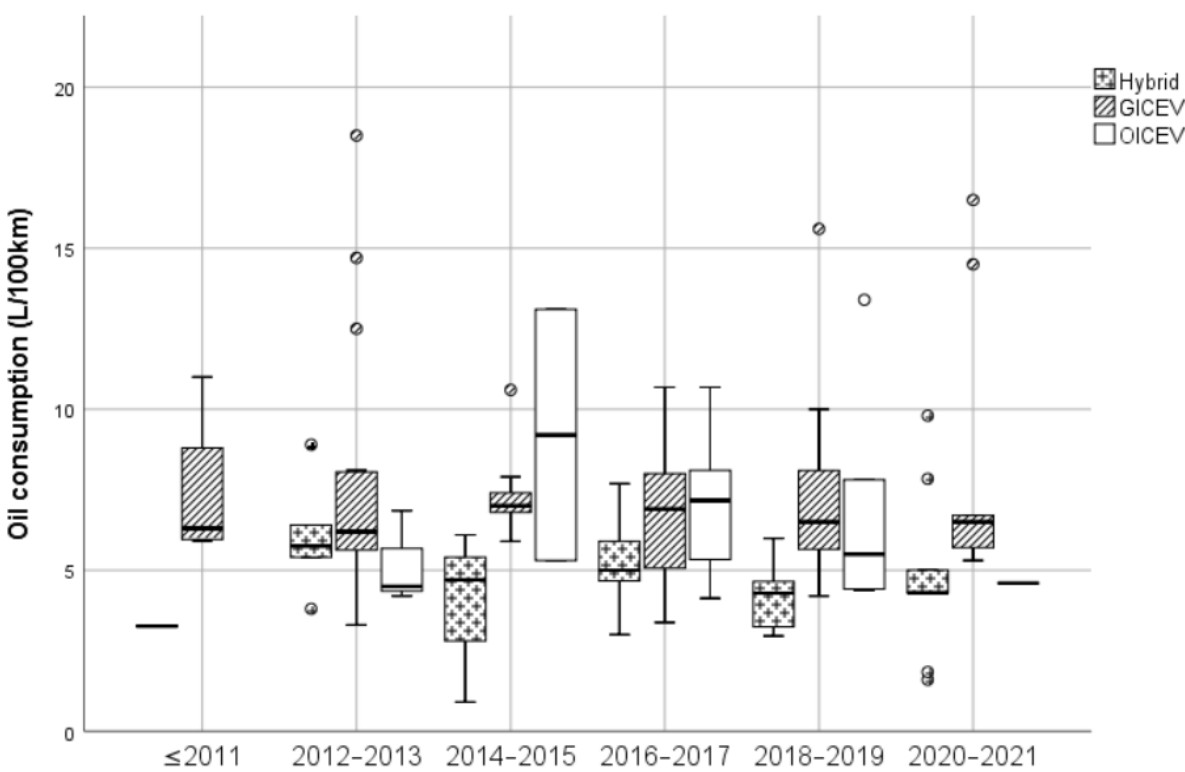

**Figure 8.** Oil consumption per kilometer.

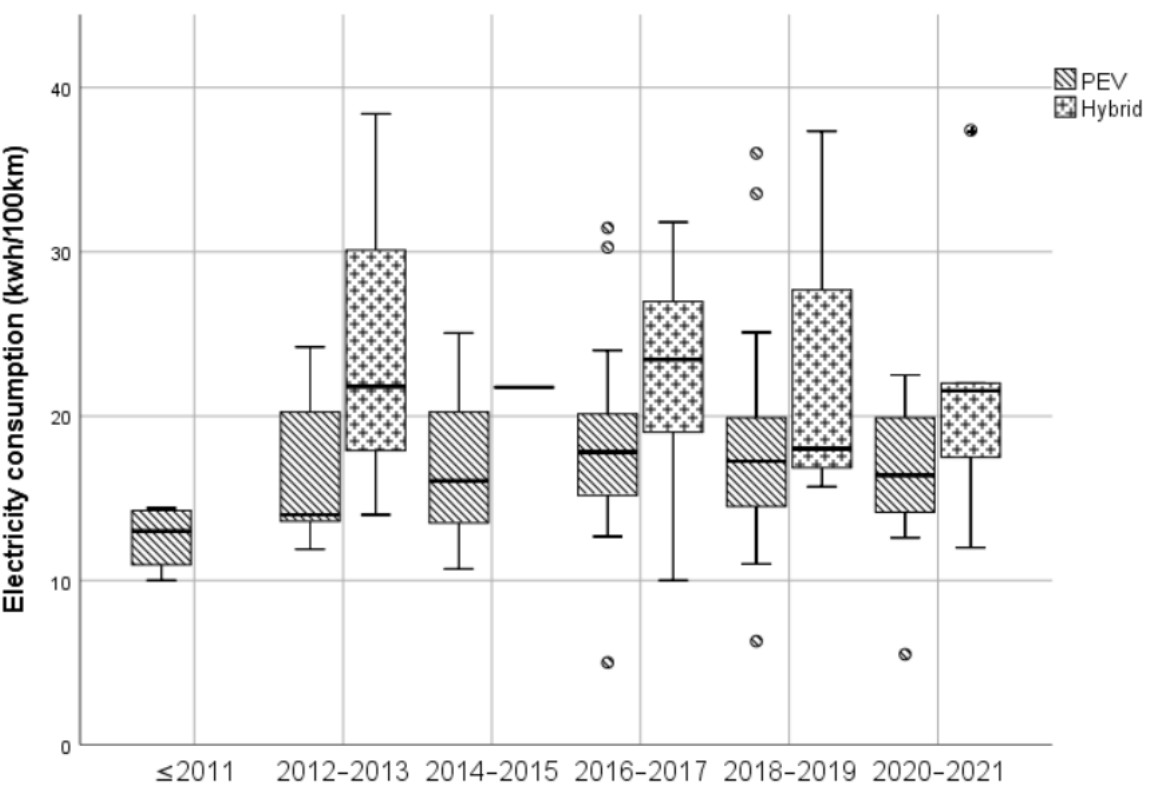

**Figure 9.** Electricity consumption per kilometer.

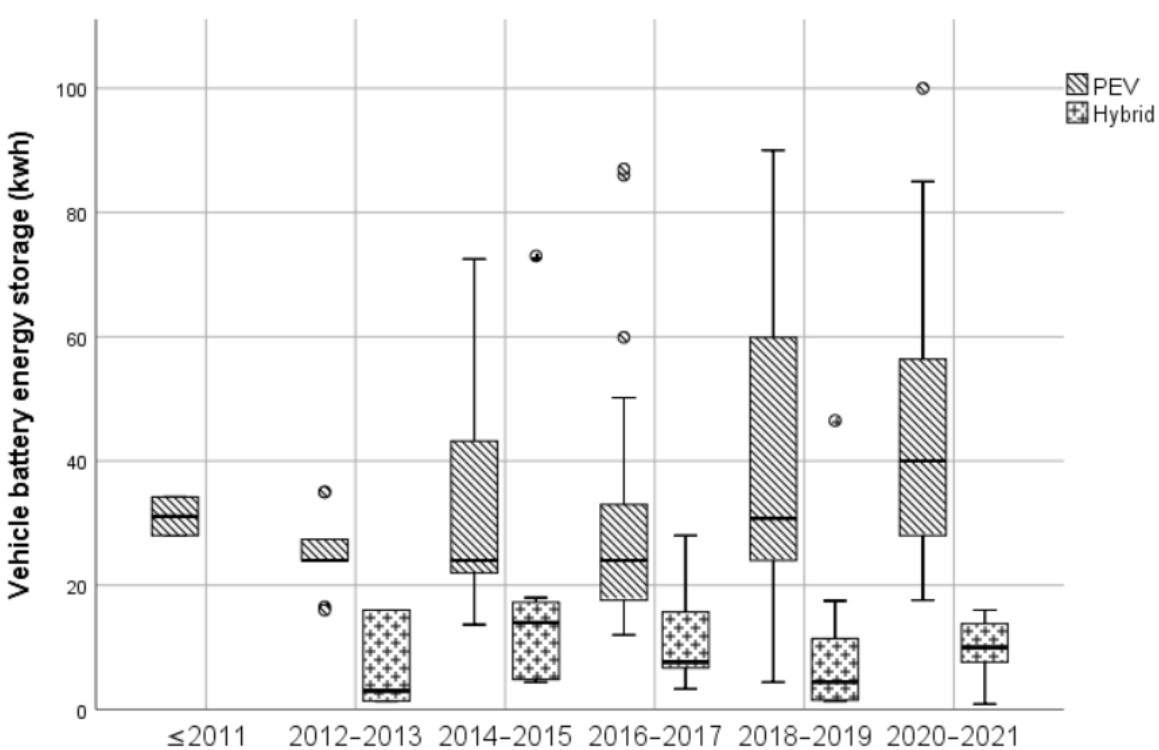

**Figure 10.** Vehicle battery energy storage.

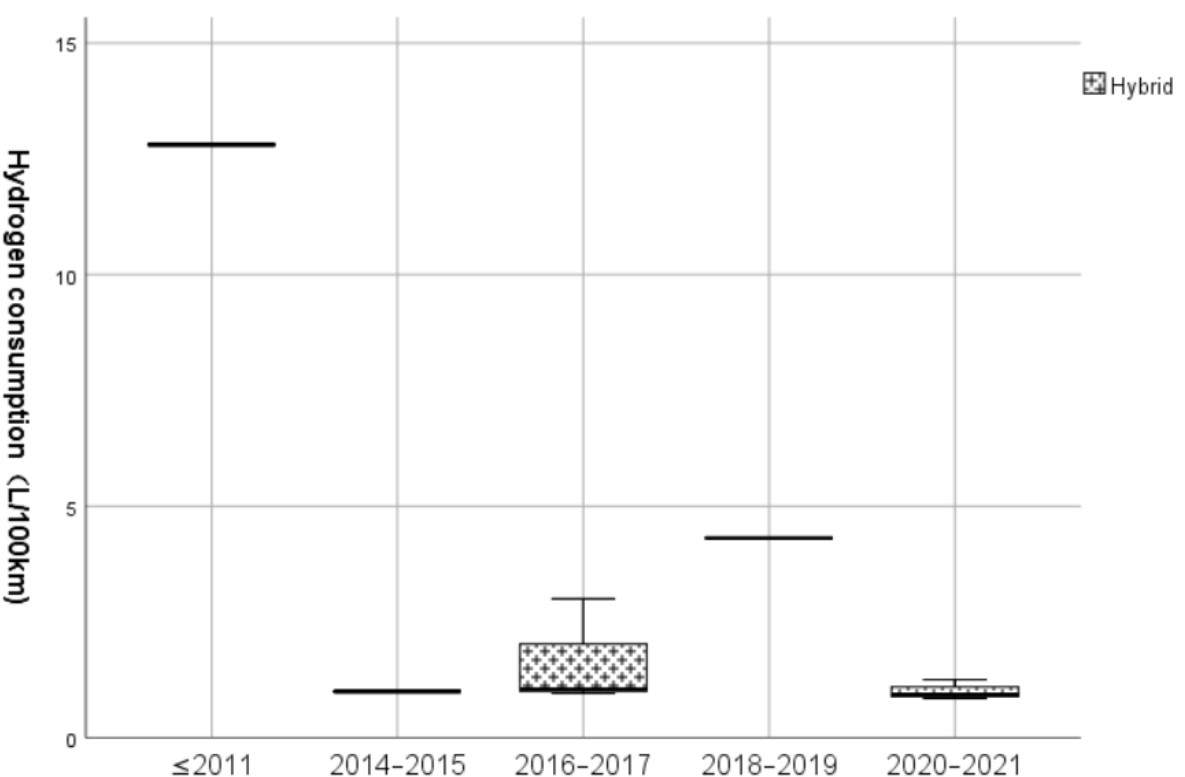

**Figure 11.** Hydrogen consumption per kilometer.

### 3.4. Environmental Efficiency Index Selection and Trends

Environmental efficiency measurement is the goal of NEV research. In addition to GWP, LCA environmental efficiency indicators include photochemical oxidant formation (POF), ozone depletion (OD), terrestrial acidification (TA), particulate matter formation

(PMF), human toxicity (HT) (including cancer and non-cancer), eutrophication (including freshwater eutrophication (FE), marine eutrophication (ME), terrestrial eutrophication (TE)), ecological toxicity (including freshwater aquatic toxicity (FT), marine aquatic toxicity (MT) and terrestrial aquatic toxicity (TT)), and land use (including rural land use, urban land use, and natural land use), ionizing radiation (IR), abiotic depletion (AD), fossil fuels depletion potential (FDP), metal depletion potential (MDP), water consumption and ozone formation potential (OFP) (including land ecology and human toxicity) and noise pollution [152], were collected in the study of the environmental efficiency scale (Table 7), Since the scale was constantly updated, the changes of versions were not counted. Excluding ISO14040/ISO14044, which is the most basic scale, the measurement scales can be divided into three types. One is a scale specifically for evaluating a certain index, including CED [153] for evaluating energy, UseTox [149] for evaluating toxicity, and HBEFA [154] for evaluating exhaust gas. The other is a scale containing multiple indicators, including ILCD, Eco-Indicator, CEENE, EPS2000, IMPACT [155], and EPD [156]. Recipe, CML, TRACI, and Environmental Footprint [139] are the evaluation scales used by Gabi. The last one is the comprehensive evaluation scale, which assigns a certain weight to selected indicators and gives comprehensive environmental efficiency, including EI99 [15]. None of these scales measured the environmental impact of noise pollution [152].

**Table 7.** Evaluation scale selection.

| Rank | Evaluation Scale | Numbers |
|---|---|---|
| 1 | ISO 14040/ISO 14044 | 66 |
| 2 | Recipe | 27 |
| 3 | CML | 24 |
| 4 | IPCC | 16 |
| 5 | ILCD | 7 |
| 6 | CED | 5 |
| 7 | EI99 | 5 |
| 8 | Eco-indicator | 4 |
| 9 | IMPACT | 4 |
| 10 | Environmental Footprint | 1 |
| 11 | CEENE | 1 |
| 12 | TRACI | 1 |
| 13 | EPS | 1 |
| 14 | UseTox | 1 |
| 15 | HBEFA | 1 |
| 16 | EPD | 1 |
| 17 | No introduction | 81 |

Selection and Trends of Vehicle Environmental Efficiency Indices

The times of using different indicators were counted to evaluate environmental efficiency (Table 8). For statistical convenience, rural land use, urban land use, and natural land use were unified as land use [157], and human toxic-carcinogenic and human toxic-noncarcinogenic were unified as human toxicity. In particular, the statistics of TA, measured in mol/km [78], FDP, measured in MJ/km [34] and MDP, measured in iron [36,158,159] were also calculated. In addition to GWP, POF, TA, PMF, HT, and FE are high-frequency selection indices.

The most widely discussed topic of the greenhouse effect and 2–6 environmental efficiency indicators for research were selected. These indicators not only contain sufficient data but are also very important for the evaluation of the environmental efficiency of NEVs [160–162]. The WTW $CO_2$ emissions of NEVs depend on the local power generation configuration level and fuel quality [10,74,88]. According to the WTW emission statistics of Figure 12, the average WTW emissions of both EVs and hybrids are lower than those of GICEVs. In 2021, the average WTW emissions of EVs were 97.01 g $CO_2$-eq/km. The average value of a hybrid is 120.68 g $CO_2$-eq/km, while that of a GICEV is 210.35 g $CO_2$-eq/km.

Due to the mining of raw materials and the production of spare parts for the batteries of NEVs, the GWP of NEVs in the production stage is significantly higher than that of ICEVs [56,136,147,153]. As seen from the production stage shown in Figure 13, the $CO_2$ emissions in the production stage of EV always maintained the highest state. In 2021, the average value of an EV is 81.78 g $CO_2$-eq/km, that of a hybrid is 65.86 g $CO_2$-eq/km and that of a GICEV is 42.87 g $CO_2$-eq/km. When $CO_2$ emissions are viewed from the perspective of the whole life cycle (as shown in Figure 14), the $CO_2$ emissions of EVs are still better than those of GICEVs but are like those of hybrids. The $CO_2$ emissions of EVs are higher than those of hybrids [8,34] or lower [130,136], resulting in no clear conclusion. In 2021, the emission rate of the average EV is 180.41 g $CO_2$-eq/km, for the average hybrid, it is 190.11 g $CO_2$-eq/km, and for the average GICEV, it is 293.55 g $CO_2$-eq/km.

In addition to GWP, POF, TA, PMF, HT and FE are all hot topics of concern. Among them, NEVs can improve the formation of POF to a certain extent (Figure 15), especially in optimizing the impact of ethanol fuel [160]. This can reduce the formation of photochemical smog by 51% on average [162]. In 2021, the average emission rate of an EV is 0.26 g NMVOC-eq/km, for the average hybrid it is 0.18 g NMVOC-eq/km, and for the average GICEV, it is 0.34 g NMVOC-eq/km.

NEVs, especially EVs, reduce environmental efficiency in terms of TA [30,38,158,163], HT [18,30,38], and FE [38,158,161,164], as shown in Figures 16–18. In 2021, the average TA, HT, and FE of EV were 1.01 g $SO_2$-eq/km, 96.80 g 1,4-DCB-eq/km, and 0.10 g P-eq/km, respectively; for a hybrid, 0.38 g $SO_2$-eq/km, 30.34 g 1,4-DCB-eq/km, and 0.04 g P-eq/km; while for GICEV, 0.79 g $SO_2$-EQ/km, 5.76 g 1,4-DCB-eq/km and 0.03 g P-eq/km, respectively. This phenomenon is mainly caused by the production of lithium batteries, namely, the production stage of NEVs [14,18].

The environmental efficiency of NEVs, in terms of PMF (Figure 19), is inconsistent with ICEVs because NEVs produce more particulate matter than ICEVs during the production stage [158,165] but less during the energy stage [14,158]. Meanwhile, OICEVs, especially diesel ICEVs, have the lowest environmental efficiency [166]. In 2021, the average for an EV is 0.09 g P2.5-eq/km, for a hybrid, it is 0.06 g P2.5-eq/km, and for a GICEV, it is 0.19 g P2.5-eq/km.

**Table 8.** Vehicle environmental efficiency index selection ranking.

| Rank | Environmental Efficiency Index | Numbers |
|:---:|:---:|:---:|
| 1 | GWP (g $CO_2$-eq/km) | 520 |
| 2 | POF (g NMVOC-eq /km) | 143 |
| 3 | TA (g $SO_2$-eq/km) | 140 |
| 4 | PMF (g PM2.5-eq/km) | 137 |
| 5 | HT (g 1,4-DCB/km) | 88 |
| 6 | FE (g P-eq/km) | 79 |
| 7 | FDP (g Oil-eq/km) | 73 |
| 8 | OD (g CFC-11-eq/km) | 70 |
| 9 | FE (g 1,4-DCB-eq/km) | 53 |
| 10 | IR (bq U235-eq /km) | 43 |
| 11 | TE (g 1,4-DCB/km) | 39 |
| 12 | Water Consumption (l/km) | 37 |
| 13 | Land Use ($m^2$/km) | 34 |
| 14 | MDP (g Cu-eq/km) | 30 |
| 15 | AD (g Sb-eq/km) | 29 |
| 16 | MT (g 1,4-DCB/km) | 27 |
| 17 | ME (g N-eq/km) | 27 |
| 18 | TE (g N-eq/km) | 16 |
| 19 | OFP-Terrestrial Ecological (g $NO_x$-eq/km) | 11 |
| 20 | ODP-Human Toxicity (g $NO_x$-eq/km) | 10 |

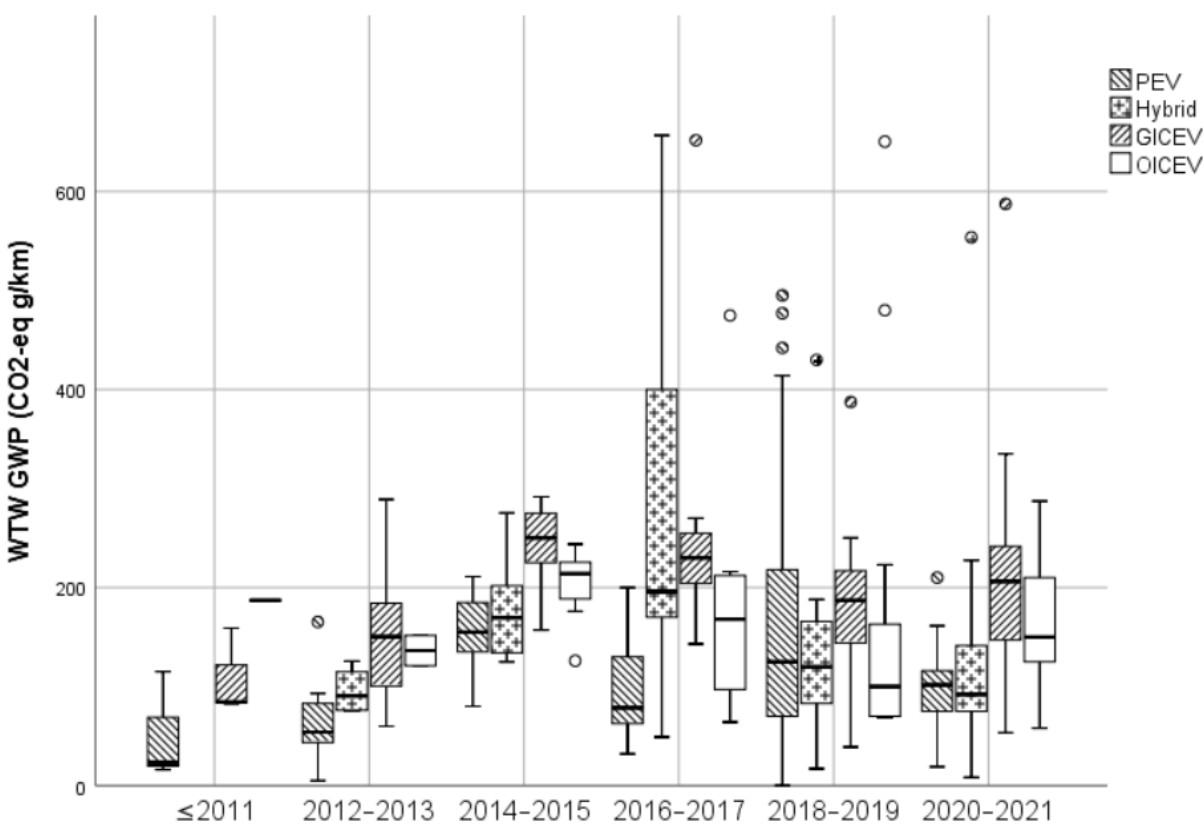

**Figure 12.** CO$_2$ emissions per kilometer (WTW).

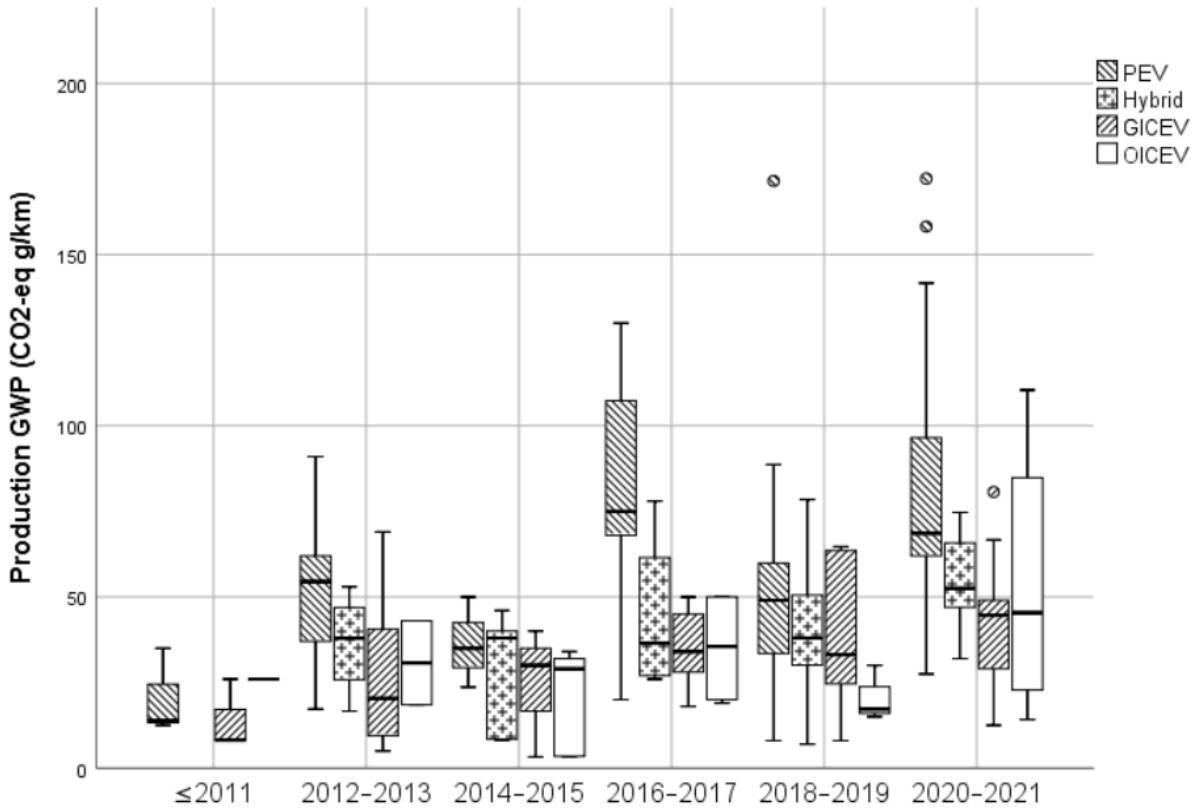

**Figure 13.** CO$_2$ emissions per kilometer (production).

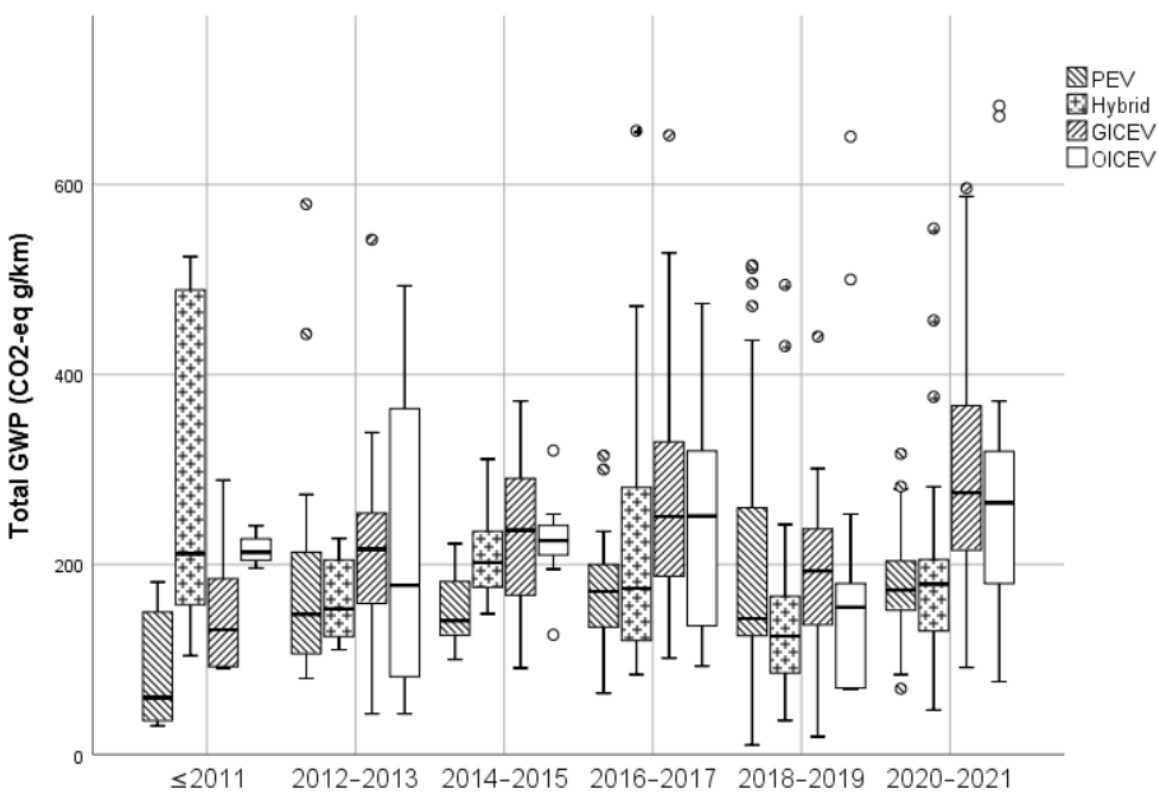

**Figure 14.** CO$_2$ emissions per kilometer (total).

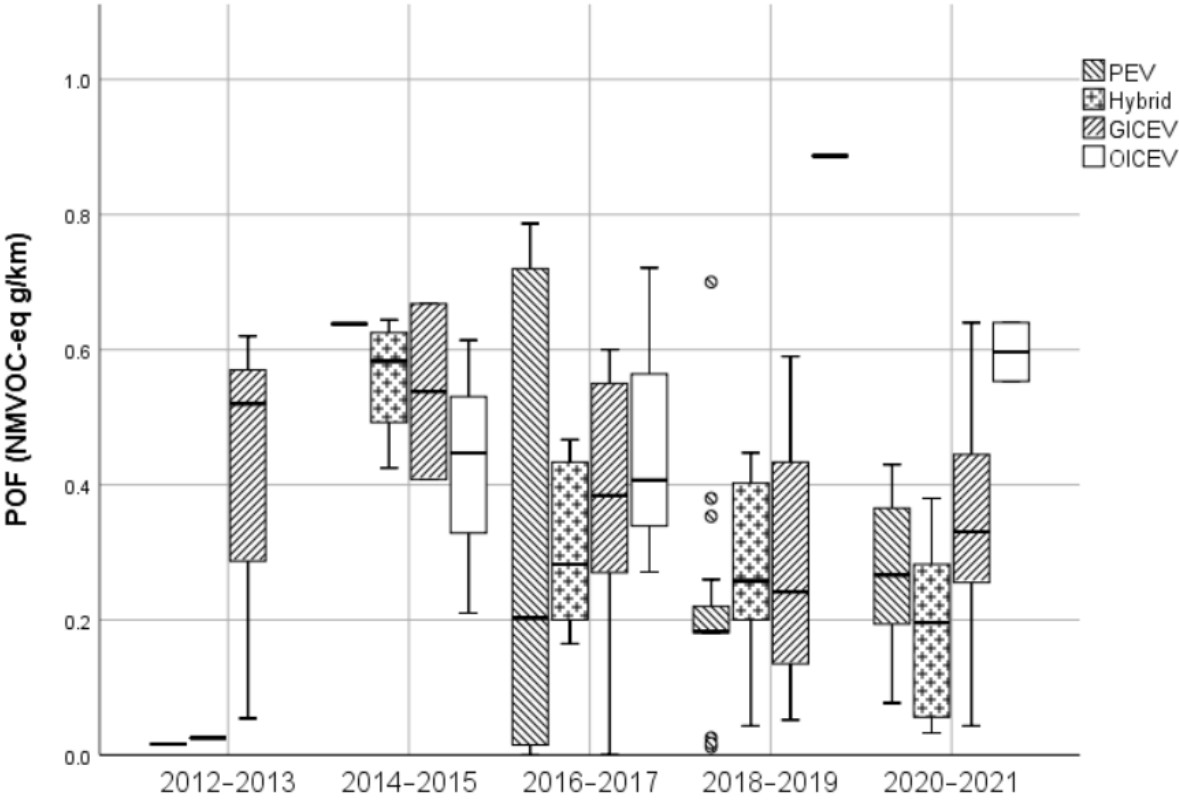

**Figure 15.** POF per kilometer (Total).

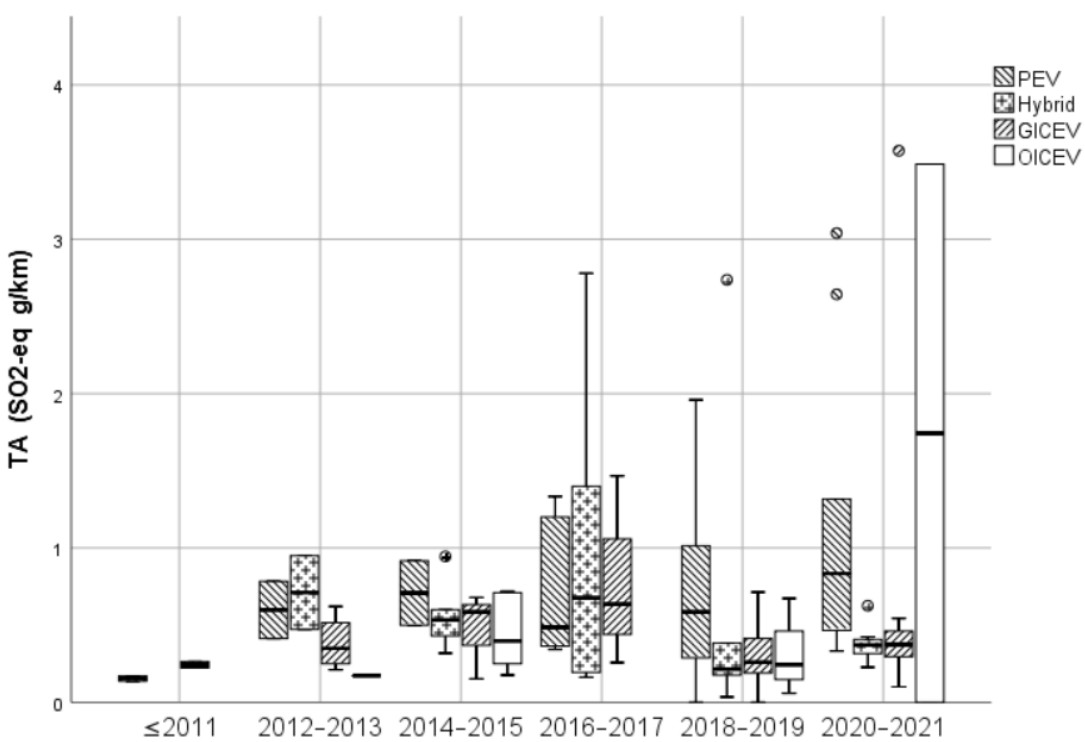

**Figure 16.** TA per kilometer (total).

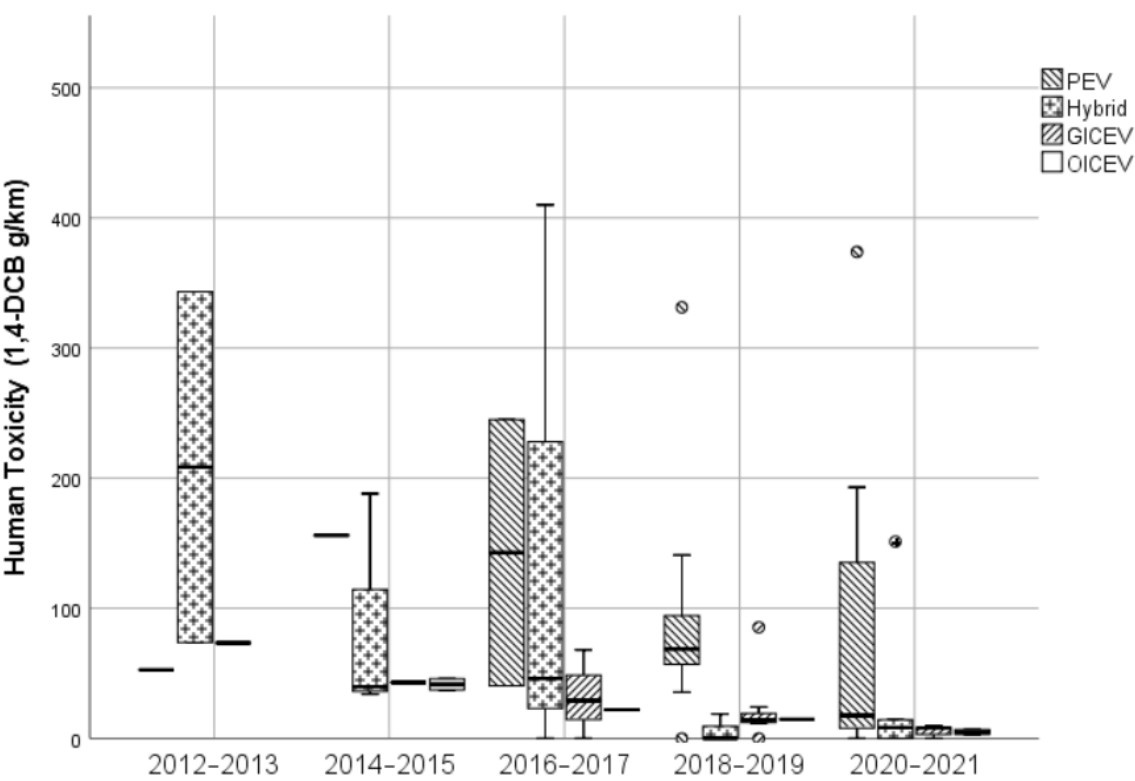

**Figure 17.** HT per kilometer (total).

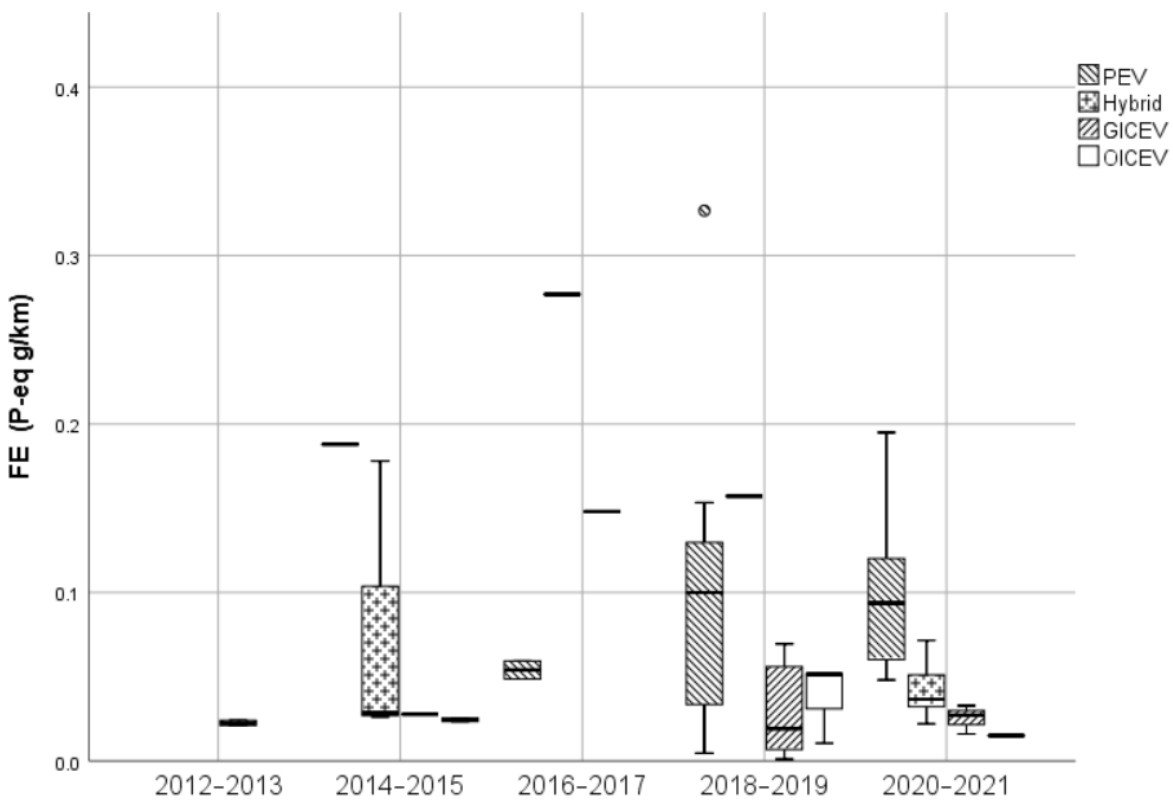

**Figure 18.** FE per kilometer (total).

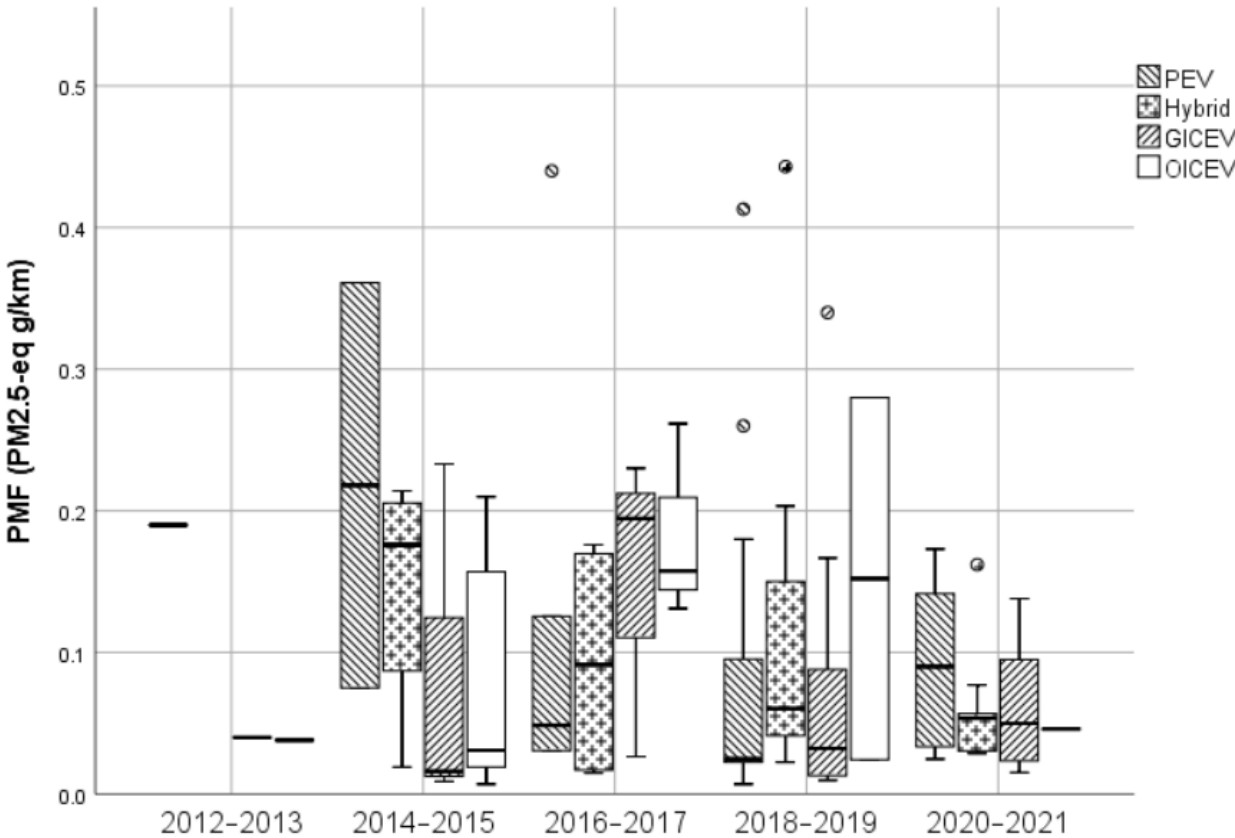

**Figure 19.** PMF emissions per kilometer (total).

## 4. Discussion

### *4.1. More Comprehensive Process*

#### 4.1.1. Materials, Equipment and Car Transportation

Like energy transportation, the raw materials, semi-finished products, components, and even the finished vehicles themselves need to be transported to a designated location to connect with the next link in the process. Auto parts need to be transferred from parts suppliers to automobile manufacturers, into whole vehicles [147,167]. This can range from the short-distance transportation of materials and semi-finished products in industrial ecological parks, for example, 7.7 km [138], to the transport of finished vehicles from a key province in a country, for example, Guangdong to Heilongjiang, China, a total of 2791.03 km [136]. Because of the unclear data on raw materials or spare parts suppliers, or on the shipment situation of electric car manufacturers, it is difficult to determine the average transport distance from suppliers to manufacturers [138]. The risks of the raw material supply chain also should be considered, especially in terms of different materials. Among these, for producing lithium batteries, the supply risk of crude oil is the lowest, while the supply risk of natural graphite, fluoride, and phosphate is gradually becoming unacceptable [107]. Raw materials or finished goods vehicle transport links are not widely considered because of the unclear information about raw materials or spare parts suppliers, or the shipment situation of electric car manufacturers, it is difficult to determine the average transport distance of the required parts from suppliers to the electric car manufacturers [138]. There is another reason that transportation is considered too small a proportion of environmental efficiency to be considered [108]; most researchers prefer to just mention transport links in the study of boundaries.

#### 4.1.2. Operation Equipment Settling and Using

Operating equipment that is built and put into use to ensure the normal operation of NEVs, such as large charging piles or power plants, can alleviate range anxiety, but building too many can be a waste of money and reduce the efficiency of the NEV environment, even if the construction of NEV operating equipment is efficient and investment environment efficiency is much higher than those for ICEVs [135].

The construction of smart grids, especially in terms of vehicle-to-grid (V2G), is the key to achieving real energy conservation and emissions reduction [168]. However, the copper demand for EV charging grid upgrades is expected to account for 74% of all copper demand growth in 2025. Considering the ripple effect of cumbersome automobile production and the supply chain, potential future system environmental changes or resource shortages could increase the cumulative amount of resource consumption and emissions by several orders of magnitude [104]. When considering the construction of the operating equipment, the disadvantages of $SO_2$ will continue to expand [169]; the weight of environmental efficiency in operating equipment seems to have become more significant in terms of the environmental efficiency as a whole.

The inability of obtaining reliable access to efficient charging stations is one of the serious obstacles to purchasing NEVs, second only to price and range [135]. Similar to electric roads, most of the emissions of charging piles come during the construction stage. For 5.29 g $CO_2$-eq/km charging pile LCA emissions, approximately 5.28 g $CO_2$-eq/km $CO_2$ emissions come from the construction stage [9]. The financial benefits obtained by laying charging piles in different regions are also diverse [35]. Furthermore, optimizing the site selection of charging piles [125,126,170] and analyzing charging transaction data [151] can further improve environmental efficiency. Under mature charging information interaction, the environmental efficiency can be optimized by 1.16~2.90% by 2030 and 0.89~5.36% by 2040 [151].

The basic NEV LCA process model was constructed by Nordelof [6,78]. After considering the production materials, parts, and construction, and putting into use vehicle transportation and operation equipment, we not only add the equipment life cycle, with material, equipment, and cars' transportation stages, to enhance the importance of time

span but also add the operation equipment stage, to emphasize the necessity of extra resource availability and efficiency [6,78]. The updated model is shown in Figure 20.

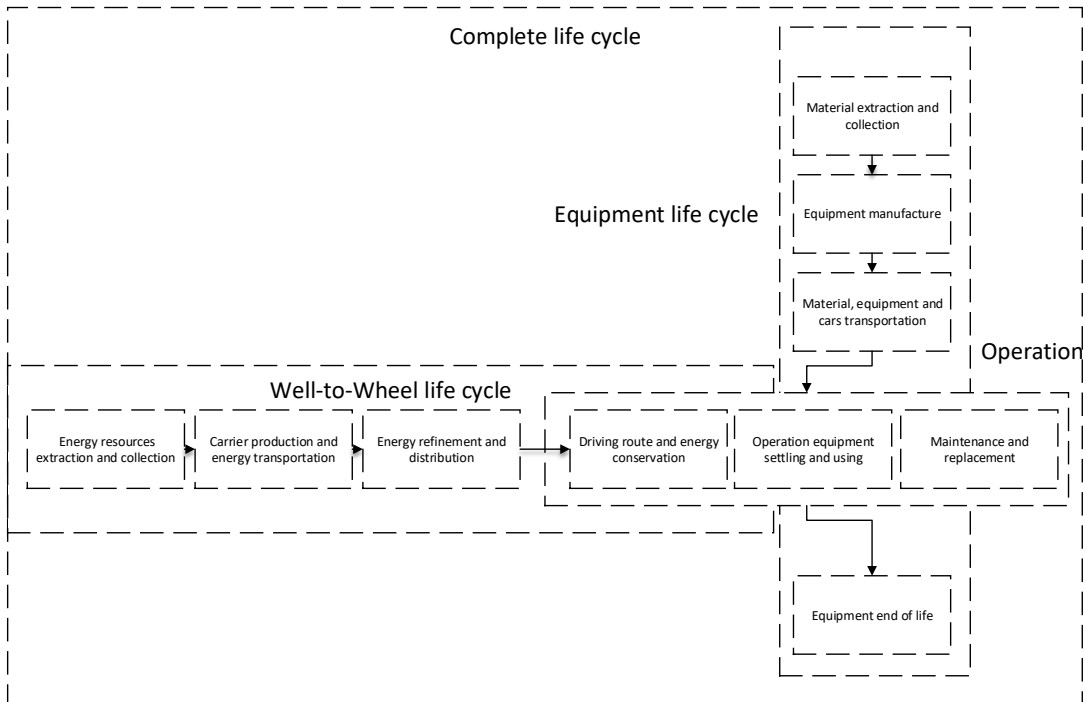

**Figure 20.** Improved basic vehicle life cycle analysis model.

*4.2. More Novel Research*

4.2.1. Environmental Assessment of HFCEV

HFCEVs also transfer the pollution caused by road driving to power stations and even hydrogen production plants. In addition, due to the highly explosive nature of hydrogen, the preparation, transportation, and use of hydrogen must pay more attention to safety. The LCA environmental assessment of HFCEVs mainly focuses on the energy stage (WTT).

Hydrogen production varies not only in terms of raw materials (naphtha cracking, steam methane reforming, electrolysis, coke oven gas purification) [83] but also in terms of energy sources. When hydrogen produced from natural gas reforming is used as a fuel, it can reduce WTW fossil energy consumption by 5–33% and WTW GWP by 15–45% [90]. However, as technology changes, the advantages of hydrogen production from natural gas gradually decrease [171], and the energy efficiency of hydrogen production from natural gas is low [172]. This makes hydrogen production from natural gas no longer stable, especially while the database is not unified, and high uncertainty and inaccuracy appear [22,90]. Clarifying the rapidly changing power structure of various countries/regions, tracking the daily, even hourly, power generation configuration changes, considering the energy required for hydrogen compression and liquefaction [90], and scientifically measuring the environmental efficiency and risk of nuclear power are all problems that need to be solved to correctly measure the environmental efficiency of HFCEVs [22].

Compared with other vehicles, the environmental efficiency of HFCEVs has gone through a process from fuzzy to relatively clear. From the beginning, it was not clear which form of methanol as an energy carrier among HEV, FCEV, HFCEV, and EV had the best environmental performance [21], from a compressed natural gas vehicle with the highest overall environmental score to a hydrogen energy battery vehicle with the lowest score. HFCEVs have high water consumption (12.02 L/km) and low energy efficiency (30.7%) [173]. Later, it was found that the average WTW $CO_2$ emissions of HFCEVs were 35% of those of ICEVs, 47% of those of HEVs, and 63% of those of EVs [83]. Finally, it was found that HFCEVs that are fueled by hydrogen and nuclear hydrogen had outstanding

potential in reducing $CO_2$ emissions (61.20–129.48 g $CO_2$-eq/km). Private passenger vehicles fueled by natural gas and hydrogen are comparable to internal combustion engine vehicles (187–235 g $CO_2$-eq/km) [52]. However, these cases all have strong regional attributes, and different power generation configurations in each region result in differences in the material cost and energy required for producing hydrogen.

### 4.2.2. Vehicle Type Classification

For the first time in history, approximately 55% of EV models on the market in 2021 will be SUVs, which is up from 45% two years ago, while the electrification rate of SUVs is about the same as that of non-SUVs. The majority of global EV sales in 2021 will still be non-SUVs, mainly due to China's preference for small EVs (such as BAIC New Energy and Wuling Hongguang) over electric SUVs (such as NiO and Ideal) [174]. The growth in the sales of SUVs is one of the main reasons for the increase in energy-related $CO_2$ emissions in 2021, and it is important to study the difference in environmental efficiency between SUVs and non-SUVs, even between different models.

In China, EVs and PHEVs of the same vehicle weight class are 24~31% GWP higher than those of GICEVs [175]. For medium and heavy vehicles, heavy EVs significantly reduce the emissions of $CO_2$, $NO_x$, volatile organic compounds, and CO in the WTW, while medium and heavy vehicles driven by diesel significantly reduce the environmental efficiency [50]. In all regions except Africa, the $CO_2$ emissions of EVs in the full-size luxury category were lower than those of gasoline ICEVs without considering database differences, while in the minicar category, Europe and Central and South America were the only regions where the $CO_2$ emissions of EVs were lower than those of GICEVs. EVs do not necessarily have a better $CO_2$ reduction effect than ICEVs, regardless of the type [159,176].

### 4.2.3. Water Footprint

Similar to the carbon footprint [97], LCA can also be used to track and calculate statistics regarding the water footprint of NEVs at various stages, and the main database used is Ecoinvent [164,177]. The adoption of EV technology has directly affected regional water consumption [178,179], so research on the water resources and water footprint of NEVs should be considered. The water footprint of NEVs is mainly discussed together with the carbon footprint [3,97,180]. The main factor of water consumption is its application in power generation, of which EVs have the highest water footprint, mainly due to upstream power generation water usage [5] and automobile and battery production [177]. EVs charged by solar energy have the lowest water consumption and can reduce the water footprint by 97%, but may consume 70 times more water than GICEVs in the worst case [145]. The huge disparity of water footprint between EVs and GICEVs needs to be clarified.

### 4.2.4. Battery Aging

Replacing parts or using aging parts to repair a car will lead to additional GWP, which will increase by 5–10 g $CO_2$-eq/km every year [159]. However, batteries will age with the rapid changes of driving operation and temperature, making range anxiety gradually rise [134] and battery efficiency continuously decrease [27]. Therefore, environmental efficiency is further reduced [148]. To some extent, ignoring battery aging overestimates the environmental efficiency of NEVs.

Battery degradation can be divided into two categories, namely, periodic degradation and calendar degradation [126]. If the battery capacitance is large enough, the calendar degradation of the battery can be desalinized [57], or loading the car with larger and higher-capacity batteries can slow down the cycle degradation and calendar degradation at the same time. With the same 160,000 km driving range, the 100-kWh battery of the Tesla S can maintain more than 93% of the factory capacity without replacing the battery. However, the 40 kWh battery of the Nissan Leaf can only maintain approximately 62% [58].

It is difficult to predict battery degradation [154], but it is one solution to simplify battery aging into battery replacement [154]. However, battery replacement does not

simply represent battery aging because battery aging is a linear exponential [128], or even an exponential process [24,126], while battery replacement is a 0–1 process. Yang, for the first time, considered both battery aging and battery replacement in terms of the LCA environmental efficiency evaluation of NEVs and pointed out that when battery aging is ignored, the $CO_2$ emissions per kilometer of NEVs will be underestimated by 29% [134].

We brought this research into our statistics, updating the WTW $CO_2$ emissions statistics for EVs and discounting the impact on PHEVs, HEVs, FCEVs, and EREVs due to the small battery size. The results are shown in Figure 21, where the up and down quartile range of EV changes from 130–249 g $CO_2$-eq/km to 143–262 g $CO_2$-eq/km. This increase makes hybrids the most environmentally efficient vehicle category in terms of GWP [75].

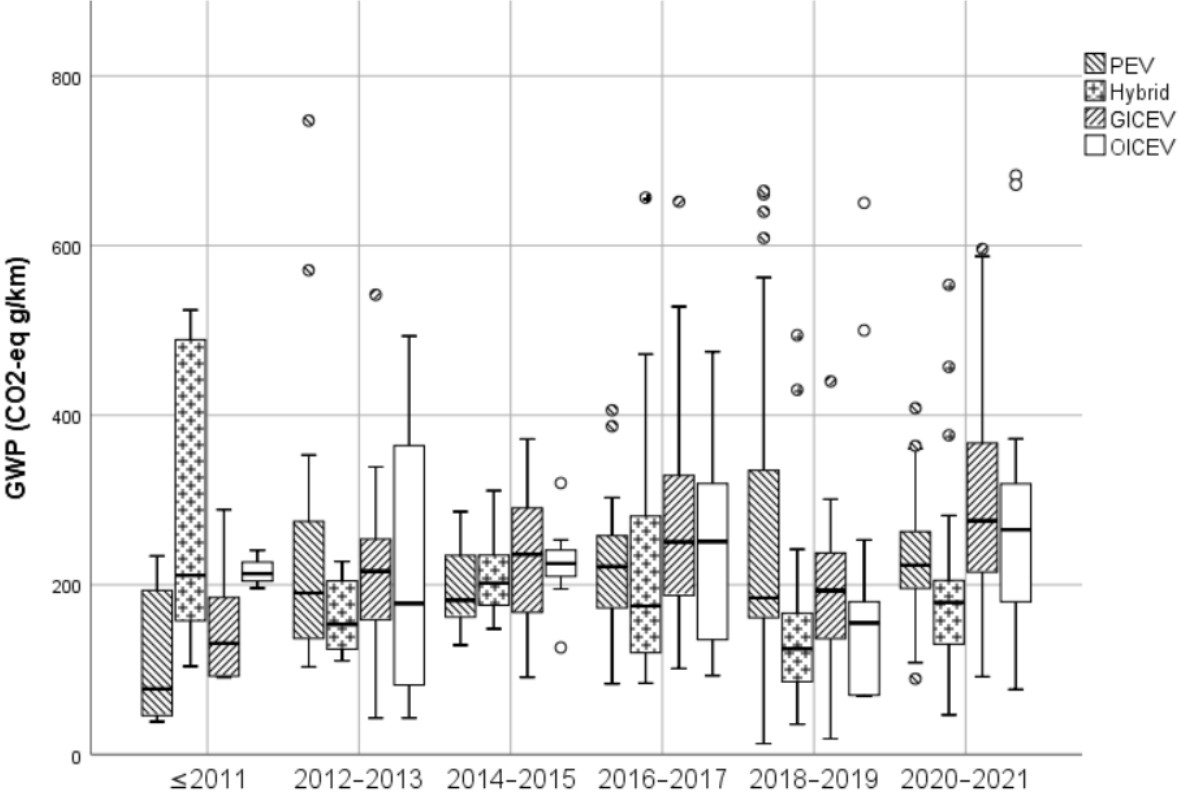

**Figure 21.** Consideration of battery age and vehicles' $CO_2$ emissions (total).

*4.3. More Diverse Evaluation*

4.3.1. Method Combination under LCA

Simple LCA research is faced with problems such as a dependence on databases and software, an inability to quantify the social impact, difficulty in simulating the driving environment, and a lack of decision-making under different conditions. Combined with different methods, LCA can better remedy the defects and more accurately reflect the environmental efficiency of NEVs.

The simulation data and scenes are substituted into LCA statistics to make the statistical data more accurate. It is a common method to use MATLAB to simulate the real situation of automobiles, which generally focuses on the dynamics model [121,153,181] of automobiles, as well as the energy composition [182] and battery system [25].

LCA needs to be combined with the corresponding models to solve decision-making, game, and uncertainty problems to diversify its role. This will assist LCA in establishing the environmental and economic efficiency of vehicles using a multicriteria decision-making method (MCDA) [26,183] or impacts related to the regional and global supply chains of the automotive industry [3], and evaluating the environmental efficiency of NEVs from an industrial perspective, using a four-quadrant matrix [184]. When the risk of the supply

chain increases [29] or the environmental rebound effect occurs [131,185], the environmental efficiency of NEVs as measured by LCA will be distorted. Combined with material flow analysis (MFA), the inventory and material changes of products and materials over time can be quantified along the value chain, tracking environmental benefits throughout the cycle's value chain and even at the end of equipment life [139].

### 4.3.2. Comprehensive Evaluation under LCA

LCA collects and creates statistics on the environmental efficiency of each environmental index at each stage of NEVs, without considering the different importance of indicators, so it cannot reflect the comprehensive environmental benefits of NEVs in the form of the total value.

The comprehensive evaluation of NEVs under LCA uses the definition of LCA to divide the stages of NEVs, determine the weight of indices, and measure the comprehensive environmental efficiency of NEVs under certain restrictions. The commonly used comprehensive evaluation methods include the analytic hierarchy process (AHP) [97,167,180], multigrade fuzzy method [186], input–output model [53,77], Spearman's partial rank correlation coefficient evaluation [95], and data envelopment analysis (DEA) [187]. A variety of similar evaluation methods are selected and combined without weight division, with the single environmental efficiency measurement expanded to the LCSA based on environmental, economic, and social life cycle impacts also being feasible [3].

### 5. Conclusions

This study systematically analyzed previously published papers using the LCA model to evaluate the environmental efficiency of NEVs. Taking a total of 282 papers on WOS as the research object, it integrated the research methods and stages used for evaluating the environmental efficiency of NEVs from the perspective of LCA. The research objects, research indicators, and research results selected by each paper were counted, and the research preferences and trends are described. It was found that in the use of the LCA model for evaluating the environmental efficiency of NEVs, energy resource extraction and collection, carrier production and energy transportation, maintenance and replacement are ill-considered research stages. Material, equipment and car transportation, operation equipment settling, and use should be considered in future work. HFCEV, vehicle type classification, water footprint, battery recovery and reuse, and battery aging are the key points to be further studied. Comprehensive evaluation, combined with more varied methods, is the direction of evaluation model optimization.

In the selection of research objects and indicators, Toyota is the most frequently appearing automobile manufacturer, and Nissan Leaf is the most frequently appearing automobile brand in the research. GWP, POF, TA, PMF, HT, and FE are high-frequency selection indices for evaluating NEV environmental efficiency.

The results of each study were collected. It was found that the average WTW $CO_2$ emissions of EVs and hybrids are always smaller than those of GICEVs in the same period. In 2021, the average WTW $CO_2$ emissions of EVs were 97.01 g $CO_2$-eq/km. The average value of a hybrid is 120.68 g $CO_2$-eq/km, while that of GICEV is 210.35 g $CO_2$-eq/km. The $CO_2$ emissions of EVs and hybrids are always higher than those of GICEVs in the same period. In 2021, the average value of EV vehicles is 81.78 g $CO_2$-eq/km, for hybrids is 65.86 g $CO_2$-eq/km, and for GICEVs is 42.87 g $CO_2$-eq/km. The $CO_2$ emissions of EVs are always lower than those of GICEVs in the same period, but there is no clear conclusion that the $CO_2$ emissions of EVs are higher or lower than those of hybrids. In 2021, in terms of emissions, the average EV is 180.41 g $CO_2$-eq/km, the average hybrid is 190.11 g $CO_2$-eq/km and the average GICEV is 293.55 g $CO_2$-eq/km.

It is believed that this study still has the following shortcomings. (1) The database only considered the research under the WOS database. (2) NEV-related data collection was manual, tedious, and time-consuming. (3) LCA itself contained the measurement of environmental efficiency, and the definition of "environmental efficiency" in the search

term would lead to inaccurate search results. (4) There was a lack of detailed statistics on battery environmental efficiency indicators, although such statistical information was less difficult to obtain.

Using LCA to evaluate the environmental efficiency of NEVs was constantly challenged by the actual situation. As the NIO, an automobile manufacturer, designs and equips charging stations exclusively for its own electric vehicles and gradually expands its scale, how will the environmental efficiency during the operation stage change? In the winter of 2021, the three provinces in northeast China had electricity problems caused by industrial and residential electricity supply difficulties. The main reasons were an unstable ratio of investment in clean energy and a shortage of fossil fuels, resulting in a power supply that cannot be guaranteed by previous thermal power during peak demand and power failure. How can the allocation of power configuration be scientifically regulated not only to ensure the energy-saving and emission reduction tasks of NEVs but also to ensure that industrial and residential electricity supplies are uninterrupted? In the winter of 2021, Europe is also facing an energy crisis, mainly due to supply disruptions caused by geopolitics. The question remains: how can the environmental efficiency of NEVs be measured under the conditions of supply disruption?

**Supplementary Materials:** The following are available online at https://www.mdpi.com/article/10.3390/su14063371/s1. File S1: Bibliography, Car indicators, Battery indicators, Power configuration, index of electric vehicle, index of hybrid vehicle, index of diesel vehicle, index of internal vehicle.

**Author Contributions:** Conceptualization and original draft preparation, G.T.; writing—review and editing, N.W. All authors have read and agreed to the published version of the manuscript.

**Funding:** This research is supported by the Key Project of National Natural Science Foundation of China under Grant 71732006, and the Major Project of the National Natural Science Foundation of China under Grant 72192830, 72192834.

**Institutional Review Board Statement:** This research requires no ethical approval.

**Informed Consent Statement:** This research involves no human study.

**Data Availability Statement:** Data sharing not applicable.

**Conflicts of Interest:** The authors declare no conflict of interest.

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
