# Peer review of "A Review on Environmental Efficiency Evaluation of New Energy Vehicles Using Life Cycle Analysis"

_sustainability, doi:10.3390/su14063371_

Round 1

Reviewer 1 Report

The paper presents a review of studies dealing with LCA of different types of vehicles.

It says that 282 papers were analysed. Why does the bibliography only comprise 186 papers then? For completeness, all 282 papers should be listed in the bibliography. If some of them turned out not to be suitable, the number 282 should be reduced accordingly.

The discussion section should be more comprehensive drawing conclusions from the results of the different studies examined.
How do you interpret the discrepancies between studies? What does future research have to address in particular?
Figure 21 presents an updated LCA process model. More information on why this was chosen should be given. The justification is quite short.

English language should be improved. Some terms or sentences sound rather strange or ambiguous. See also my detailed comments on some issues regarding the language.

Specific comments:

  • The last sentence of the abstract is way too long and therefore hard to understand. Try to split it into three or four sentences.
  • Line 36: "CO2 emissions will still maintain an annual growth of 1.9% by 2020" - Future ("will") is used for something that occurred in the past (2020).
  • Line 48: "media propaganda" - In liberal countries, the media usually don't distribute propaganda. Or do you refer to illiberal countries?
  • Line 96: "... when the paper begins to study". What does that mean? And when is the "beginning of the database"? 1996? Please state it.
  • Line 98: What does "removal of LCA" mean?
  • Table 2 is not referenced in the text.
  • Line 236: Are the values taken from [80]? This is unclear.
  • Line 354: Not everywhere in the world, the electricity price during the day is 2.5 as high as in the middle peak period and 3.3 times as high during the off-peak period. Which place are you referring to?

Author Response

Thank you so much for the opportunity to revise our manuscript. Your suggestions are very helpful, enlightening, and precise for improving our paper. We have studied your comments carefully and have revised relevant parts. For your convenience, all the revisions made to the manuscript are highlighted in bright blue. Our responses to your comments are as following:

  1. Number of bibliographies

It says that 282 papers were analysed. Why does the bibliography only comprise 186 papers then? For completeness, all 282 papers should be listed in the bibliography. If some of them turned out not to be suitable, the number 282 should be reduced accordingly.

Response: Thank you for your comments. We have well considered the information we want to show from the bibliographies. If we displayed all literature we searched before, the word count of this paper would increase about 30%. So, we make an eclectic solution, we display the details (mainly topic and DOIs) of all literatures in support material 1 and mention this message in line108-109.

  1. Further discussions

The discussion section should be more comprehensive drawing conclusions from the results of the different studies examined.

How do you interpret the discrepancies between studies? What does future research have to address in particular?

Response: Thank you for your comments. We have well reviewed the information that we have showed and want to show. We add some discussion about the discrepancies and future research in line599-605, line 623-625, line 659-665, line 685-688 and line 709-712.

  1. More explanation of specific figure

Figure 21 presents an updated LCA process model. More information on why this was chosen should be given. The justification is quite short.

Response: Thank you for your comments. We have well considered the information we want to show from Figure 21 and replaced a bigger and clear version. We added some explanation about refreshing basic model in line 637-642 (in the newest version Figure 21 is Figure 20).

  1. Polish of language

English language should be improved. Some terms or sentences sound rather strange or ambiguous. See also my detailed comments on some issues regarding the language.

The last sentence of the abstract is way too long and therefore hard to understand. Try to split it into three or four sentences.

Line 36: "CO2 emissions will still maintain an annual growth of 1.9% by 2020" - Future ("will") is used for something that occurred in the past (2020).

Line 48: "media propaganda" - In liberal countries, the media usually don't distribute propaganda. Or do you refer to illiberal countries?

Line 96: "... when the paper begins to study". What does that mean? And when is the "beginning of the database"? 1996? Please state it.

Line 98: What does "removal of LCA" mean?

Table 2 is not referenced in the text.

Line 236: Are the values taken from [80]? This is unclear.

Line 354: Not everywhere in the world, the electricity price during the day is 2.5 as high as in the middle peak period and 3.3 times as high during the off-peak period. Which place are you referring to?

Response: Thank you for your very detailed comments. We have well considered the information we want to display, and fixed the misunderstanding occurred in each sentence.

We split the last sentence in abstract into three sentences in line22-29; The word ”will” was replaced by “was” in line 36; We deleted the word “media propaganda” in line 48 to reduce controversy; We polished the sentence in line 96; We replace the sentence ”removal of LCA” into “Except the searching keyword of LCA” in line 98; Table 2 is referenced now in line 208(in the newest version Table 2 is Table 3); After our double checking, the values is taken from the abstract of [80], we replaced the citation to solve the misunderstanding in line 236; The story in [122] is told in German, so we added the explanation in line 354.

Special thanks to you for your enlightening comments!

Reviewer 2 Report

This paper deals with an interesting topic. EVs and similar new powertrains are considered zero pipe pollution. However, it is clear that they can be considered only in a local way as if the global way is considered they are not zero pipe pollution unless renewable energy is used. As the authors stated, EVs are considered not pollutant based on the life cycle.

The paper is too long with some sections that, as far as I'm concerned, describe concepts well known such as 3.2.3, 3.2.2.2, etc.  I start to see the discussion after 24 pages. 

Line 4.1.1 The risk of the raw material supply chain should be considered when transportation links are added to the environmental efficiency evaluation of NEVs. That is well known even with traditional powertrains. The suppliers are chosen aiming at being located as close as possible to the production plant.

Line 601.  The promotion of NEVs cannot be separated from the construction of smart grids and the large-scale operation of charging piles or changing stations. It is a well-known statement. EVs are linked to several facilities or technologies such as vehicle-to-grid, vehicle-to-home and vehicle-to-X. For many of them, smart grids are essential facilities.

Line 608. Strengthening NEV, new energy and the relations between and among the smart grid can accelerate the construction of smart grids and promote the development and application of clean energy to achieve real energy conservation, and emissions reduction is the key[169], especially vehicle to grid (V2G). However, smart grid upgrades require a large 611 amount of copper. Well known concept.

I can continue to add more paragraphs to this review. My main criticism is that is too long without summing up the most interesting topics after having analyzed more than 200 papers. The presentation should be improved.

Author Response

Thank you so much for the opportunity to revise our manuscript. Your suggestions are very helpful, enlightening, and precise for improving our paper. We have studied your comments carefully and have revised relevant parts. For your convenience, all the revisions made to the manuscript are highlighted in bright blue. Our responses to your comments are as following:

  1. Simplify concepts

The paper is too long with some sections that, as far as I'm concerned, describe concepts well known such as 3.2.3, 3.2.2.2, etc.  I start to see the discussion after 24 pages.

Response: Thank you for your comments. This literature was 14286 words in the initial version. After 4 times of polish, the number of words in the literature has cut into 9657 and we are still trying to polish it. In the newest version, we delete the section 2.5 first because we think the discussion about battery used in vehicle is kind of irrelevant with this topic. We also simplified some sentence to reduce the number of words and maintain the analysis we want to show in the meantime.

  1. Polish of language

Line 4.1.1 The risk of the raw material supply chain should be considered when transportation links are added to the environmental efficiency evaluation of NEVs. That is well known even with traditional powertrains. The suppliers are chosen aiming at being located as close as possible to the production plant.

Line 601.  The promotion of NEVs cannot be separated from the construction of smart grids and the large-scale operation of charging piles or changing stations. It is a well-known statement. EVs are linked to several facilities or technologies such as vehicle-to-grid, vehicle-to-home and vehicle-to-X. For many of them, smart grids are essential facilities.

Line 608. Strengthening NEV, new energy and the relations between and among the smart grid can accelerate the construction of smart grids and promote the development and application of clean energy to achieve real energy conservation, and emissions reduction is the key[169], especially vehicle to grid (V2G). However, smart grid upgrades require a large 611 amount of copper. Well known concept.

Response: Thank you for your comments. We have well considered the information we want to share and reduce the words of each sentences. Now the sentence in section, line 587 has changed into” The risk of the raw material supply chain also should be considered especially within different material “; The sentence in line 601 “ The promotion of NEVs cannot be separated from the construction of smart grids and the large-scale operation of charging piles or changing stations” was deleted; The sentence in line 608 has changed into ” The construction of smart grids ,especially vehicle to grid (V2G), is the key to achieve real energy conservation and emissions reduction “.

  1. More Concise statement

I can continue to add more paragraphs to this review. My main criticism is that is too long without summing up the most interesting topics after having analyzed more than 200 papers. The presentation should be improved.

Response: Thank you for your comments. We have double checked the words and sentences used in each section, especially in section 3 and 4. Here is our modification:

  1. In the section 2.2, the sentence “There is a new basis for environmental evaluation using LCA, and the combination of LCA and NEVs to evaluate environmental efficiency has gradually attracted attention” replaced by “The combination of LCA and NEVs to evaluate environmental efficiency has gradually attracted attention”
  2. In the section 3.1, the sentence “LCC follows the life cycle concept of LCA and conducts statistics and analysis on the cost involved in the whole life cycle of NEVs” replaced by “LCC follows the life cycle concept of LCA and conducts statistics and analysis on the cost instead of environment”
  3. In the section 3.1, the sentence “In addition to the environment and cost, the life-cycle impact on society is of equal concern.” was deleted.
  4. In the section 3.2.1.1, the sentence “Changes in production processes and energy sources required for production lead to differences in different energy conversion” replaced by “The combination of LCA and NEVs to evaluate environmental efficiency has gradually attracted attention”
  5. In the section 3.2.1.3, the sentence “Solar radiation is low or nonexistent early in the morning and night, and the amount and intensity of solar radiation in winter are lower than those in summer. The strength of the monsoon also affects the power structure[7], making the share of solar energy and wind energy very small in winter (down from 33.5% in summer to 5.1%)[95]” replaced by “Because of the solar radiation is highly influenced by daytime and season with the wind power is influenced by monsoon[7], the share of solar energy and wind energy can down from 33.5% in summer to 5.1% in winter[95]”
  6. In the section 3.4.1, the sentence “. In regions with a larger ratio of clean energy, the promotion of NEVs can reduce CO2 emissions more significantly, while in regions dominated by thermal power plants, it is not obvious” was deleted.
  7. In section 4.1.2, the sentence “However, smart grid upgrades require a large amount of copper. Copper demand for EV charging grid upgrades is expected to account for 74% of all copper demand growth in 2025, equivalent to 55,000 tons per year” replaced by “Copper demand for EV charging grid upgrades is expected to account for 74% of all copper demand growth in 2025”.
  8. In the section 4.1.2, the sentence “The availability of charging piles significantly affects consumers' car choices, which is reflected in the total mileage of NEVs” was deleted.
  9. In the section 4.2.1, the sentence “Similar to EVs, HFCEVs are not ideal vehicles with zero pollution and zero emissions. They also transfer the pollution caused by road driving to power stations and even hydrogen production plants” replaced by “HFCEVs also transfer the pollution caused by road driving to power stations and even hydrogen production plants”
  10. In the section 4.2.2, the sentence “The environmental efficiency of NEVs is divided according to types and statistics. Under the current power generation configuration in China, EVs and PHEVs of the same vehicle weight class are 24% ~ 31% GWP higher than those of GICEVs[174]” replaced by “In China, EVs and PHEVs of the same vehicle weight class are 24% ~ 31% GWP higher than those of GICEVs in 2019[174]”
  11. In the section 4.3.1, the sentence “LCA is more used as a statistical tool to measure environmental efficiency, and it needs to be combined with corresponding models to solve decision-making, game and uncertainty problems to diversify its role” replaced by “LCA needs to be combined with corresponding models to solve decision-making, game and uncertainty problems to diversify its role.”
  12. In the section 4.3.2, the sentence “Comprehensive evaluation requires assigning weight to each environmental indicator.” was deleted.

The number of words in the newest version has be cut down to 9445, we also added some discussion about the discrepancies and future research in section4.1.1, 4.1.2, 4.2.1, 4.2.2 and 4.2.3.

Special thanks to you for your enlightening comments!

Reviewer 3 Report

> paper shaped out well, need to modify the into with research contributions for easy understanding.

> Ref should be deleted and added with relevant as mentioned below.

Mopidevi, S., Narasipuram, R.P., Aemalla, S.R. and Rajan, H. (in press) ‘E-mobility: impacts and analysis of future transportation electrification market in economic, renewable energy and infrastructure perspective’, Int. J. Powertrains.

Narasipuram, R.P.; Mopidevi, S. A technological overview & design considerations for developing electric vehicle charging stations. J. Energy Storage 202143, 103225

> Range estimation should be more clear and it should be in real values.

> All figures should be more clear and more over Figure 21 is not clear enough.

Author Response

Thank you so much for the opportunity to revise our manuscript. Your suggestions are very helpful, enlightening, and precise for improving our paper. We have studied your comments carefully and have revised relevant parts. For your convenience, all the revisions made to the manuscript are highlighted in bright blue. Our responses to your comments are as following:

  1. Polish of language

Paper shaped out well, need to modify the into with research contributions for easy understanding.

Response: Thank you for your comments. We have well considered the information we want to show and double check the words and sentences used in the literature. The sentences marked is the modification in the newest version.

  1. Richer literatures

Ref should be deleted and added with relevant as mentioned below.

Mopidevi, S., Narasipuram, R.P., Aemalla, S.R. and Rajan, H. (in press) ‘E-mobility: impacts and analysis of future transportation electrification market in economic, renewable energy and infrastructure perspective’, Int. J. Powertrains.

Narasipuram, R.P.; Mopidevi, S. A technological overview & design considerations for developing electric vehicle charging stations. J. Energy Storage 2021, 43, 103225

Response: Thank you for your comments. We have well reviewed the literatures mentioned above and added citations in these sentences: In section 4.1.2, the article edited by Mopidevi, S is cited in sentence” The construction of smart grids, especially vehicle to grid (V2G), is the key to achieve real energy conservation and emissions reduction [169]”; Similarly, in section 4.1.2 the article edited by Narasipuram, R.P. is cited in sentence:” Furthermore, optimizing the site selection of charging piles [171]”.

  1. More specific range

Range estimation should be more clear and it should be in real values.

Response: Thank you for your comments. We have well considered the suggestion of revision and the following modifications have been made:

  1. We unified the decimal places in section 3.4.1, especially in the discussion about NMVOC. The sentence “In 2021, the average EV is 0.258 g NMVOC-eq/km, the average Hybrid is 0.177 g NMVOC-eq/km, and the average GICEV is 0.341 g NMVOC-eq/km.” was replaced by “In 2021, the average EV is 0.26 g NMVOC-eq/km, the average Hybrid is 0.18 g NMVOC-eq/km, and the average GICEV is 0.34 g NMVOC-eq/km.”
  2. We double checked the extreme number showed in each figure and once again we excluded some of the extreme numbers, trying to make range estimation clearer. We think that placing the specific number on each box figure will make figures more difficult to identify, and readers can find specific numbers in the support material 1. So, we make an eclectic solution, we use the grid to make figures more easily to be identified. We used the grid in all box figures.

  1. Clearer figures

All figures should be more clear and more over Figure 21 is not clear enough.

Response: Thank you for your comments. We have well considered the information we want to show from the pictures and replaced all figures into a bigger and clearer version (in the newest version Figure 21 is Figure 20).

Special thanks to you for your enlightening comments!

Round 2

Reviewer 1 Report

Thank you for incorporating my comments.
The paper improved a lot.
I recommend that a native English speaker proofreads the paper and makes corrections where appropriate.

Reviewer 2 Report

Dear authors,

my remarks have been addressed.